EMBO
Molecular Medicine

# Mitochondrial damage drives T-cell immunometabolic paralysis after major surgery

Simon Hirschberger [1,2], Martin B Müller[1,2], Hannah Mascolo[1], Melissa Seitz [1], Stefan Nibler[1], David Effinger [1,2], Kun Lu [3,4], Joscha Büch[3], Martin Bender [2], Tobias Kammerer[1], Sven Peterß[3], Karin Kleigrewe[5], Miriam Abele [5], Teresa Barth[6], Olga Kushnir[1,2], Axel Imhof [6], Steffen Dietzel [7,8], Bernd Wegener[9], Ralf Sowa [10], Frank Vogel[11], Peter Lamm[11], Roland Tomasi [2], Kristian Unger[12], Markus Sperandio [8], Erich Kilger[2], Simone Kreth [1,2 ✉] & Max Hübner [1,2]

## Abstract

**Cytotoxic T cell (CTL) dysfunction is a hallmark of immune paralysis after major surgery, increasing susceptibility to severe nosocomial infections and contributing to mortality in critically ill patients. The mechanisms remain poorly understood. We demonstrate that reactive oxygen species (ROS) released by myeloid-derived suppressor cells (MDSC) transiently emerging after surgery, drive perioperative CTL immunoparalysis. These ROS damage CTL mitochondria, triggering secondary mitochondrial ROS amplification and overwhelming antioxidant defenses. The resulting oxidative cascade impairs oxidative phosphorylation and suppresses CTL effector function. Concurrently, stress-induced mitochondrial hyperfusion disrupts fission-dependent translocation to the immunological synapse, exacerbating bioenergetic failure. MitoTEMPO, a mitochondria-targeted antioxidant, partially mitigates these effects, highlighting mitochondrial stabilization as a potential strategy to prevent perioperative immune dysfunction.**

**Keywords** Cytotoxic T cells; Mitochondria; Immunometabolism; Major Surgery; Immunosuppression
**Subject Categories** Immunology; Metabolism; Organelles

## Introduction

Modern surgery has undergone significant advancement in recent decades, becoming a highly specialized and effective medical tool that benefits millions of patients annually. However, its success is still significantly compromised by postoperative complications, largely attributable to immunological dysregulation (Dobson, 2020). Surgical procedures inevitably cause tissue trauma, disrupting the body's natural defenses and triggering the release of damage-associated molecular patterns (DAMPs) from injured cells (Relja and Land, 2020). These DAMPs activate pattern recognition receptors (PRRs) on immune cells, initiating an inflammatory response. Both the acute tissue damage and the resulting immunological imbalance can lead to a systemic inflammatory response syndrome (SIRS) (Vourc'h et al, 2018). This condition not only contributes to postoperative morbidity and mortality but also promotes a prolonged state of immunosuppression, which is characterized by impaired immune responses and increased vulnerability to infections (Papadopoulos et al, 2017; Xiao et al, 2011; Margraf et al, 2020). Consequently, the postoperative immune system becomes less capable of combating future challenges, which hampers recovery and increases the risk of complications for the patient (Rankin et al, 2011; Cornell et al, 2012). This is highlighted by nosocomial infection rates ranging from 25 to 35% after major surgery (Hadaya et al, 2022; Jannasch et al, 2015). Indeed, this immunosuppressive state is believed to be a leading cause of death among critically ill patients (Boomer et al, 2011; Bain et al, 2023). The mechanisms underlying perioperative immune dysfunction remain largely unclear, and currently, there are no reliable predictive markers or therapeutic options available.

T-cell immune dysfunction is a hallmark of prolonged perioperative immunosuppression (postoperative immunosuppression of T cells, PITC), with human cytotoxic CD8[+] T cells (CTL) being particularly susceptible (Franke et al, 2006; Markewitz et al, 1996, 1993; Albertsmeier et al, 2015; Sun et al, 2017; Ananth et al, 2016; Menges et al, 2012; Torrance et al, 2016; Pérez-Granda et al, 2024). As a result, these cells exhibit suppressed interferon signaling, impaired cytokine production, and reduced synthesis of cytotoxic proteins (Bain et al, 2023; Menges et al, 2012; Torrance et al, 2018; Ananth et al, 2016; Hotchkiss et al, 2013). This T-cell

[1]Walter Brendel Center of Experimental Medicine, Research Unit Immune Function and Immune Metabolism, Ludwig-Maximilians-University (LMU), Munich, Germany. [2]Department of Anaesthesiology, LMU University Hospital, LMU, Munich, Germany. [3]Department of Cardiac Surgery, LMU University Hospital, LMU, Munich, Germany. [4]DZHK (German Center for Cardiovascular Research), Partner Site Munich Heart Alliance, Munich, Germany. [5]Bavarian Center for Biomolecular Mass Spectrometry, TUM School of Life Sciences, Technical University of Munich, Freising, Germany. [6]Protein Analysis Unit, BioMedical Center, Faculty of Medicine, LMU, Martinsried, Germany. [7]Core Facility Bioimaging and Walter Brendel Center of Experimental Medicine, Biomedical Center, LMU, Munich, Germany. [8]Institute of Cardiovascular Physiology and Pathophysiology, Walter Brendel Center for Experimental Medicine, LMU, Munich, Germany. [9]Department of Orthopaedic Surgery, Physical Medicine and Rehabilitation, LMU University Hospital, LMU, Munich, Germany. [10]Department of Anesthesiology and Intensive Care Medicine, University Medical Center Goettingen, Goettingen, Germany. [11]Artemed Hospital Munich South, Munich, Germany. [12]Department of Radiotherapy and Radiooncology, LMU University Hospital, LMU, Munich, Germany. ✉E-mail: simone.kreth@med.uni-muenchen.de

immunoparalysis results in an inability to effectively clear pathogens and an increased vulnerability to secondary infections (Huang et al, 2022; Jensen et al, 2018). Despite these well-documented consequences, the mechanisms driving T-cell dysfunction remain poorly understood. Given the critical importance of perioperative immunological complications, it is essential to deepen our understanding of this phenomenon.

We employed a multi-omics screening approach to identify the mechanisms underlying CTL immunosuppression following major surgery. Our data show that perioperative T-cell dysfunction is linked to systemic oxidative stress induced by transiently emerging myeloid-derived suppressor cells (MDSC). Functional mitochondrial analyses revealed that ROS released by MDSC increase ROS levels in CTL, thereby compromising mitochondrial integrity, inducing mitochondrial hyperfusion and impairing mitochondrial translocation to the immunological synapse (IS), ultimately leading to bioenergetic dysfunction of CTL.

# Results

## Post surgical immunosuppression (PITC) of CTL after major surgery

Immune-related postoperative complications significantly impact patient outcomes. To assess postoperative immunosuppression following major surgery, we enrolled patients scheduled for cardiothoracic (aortocoronary bypass, aortic arch replacement) and major orthopedic procedures (polysegmental spinal stabilization) and analyzed CTL at three time points (Fig. 1A): preoperatively (T1), postoperatively (T2), and on the first day after surgery (T3). Patients with preexisting conditions associated with immune dysregulation, such as cancer, autoimmune diseases, immunosuppressive therapy or active infections were excluded from the study.

At T2, CTL exhibited a dysfunctional phenotype. After T-cell-specific stimulation, transcript levels of interferon gamma (IFNG), granzyme B (GZMB), and perforin 1 (PRF1) mRNA were substantially reduced (Fig. 1B). Intracellular IFN-γ protein expression was significantly decreased, and IFN-γ secretion was almost completely abrogated (Fig. 1C,D). We next evaluated the cytotoxic function of CTL using Electric Cell-Substrate Impedance Sensing (ECIS) and a flow-cytometry based killing assay, revealing a significantly impaired cell lysis capacity of T2 CTL (Figs. 1E and EV1A). To assess whether the observed functional impairments were also reflected in classical activation and exhaustion markers, we compared their expression at time points T1 and T2. As expected — given the short time interval — canonical exhaustion markers remained unchanged. (Fig. EV1B). Similarly, the activation markers CD25 and CD69 showed no differences between time points (Fig. EV1B). Notably, only CD38 was significantly upregulated following major surgery (Fig. 1F), suggesting a possible link to impaired cellular metabolism (Ghosh et al, 2023).

Consistent with their reduced cytolytic capacity, T2 CTL also induced lower caspase activity in K562 cells, also suggesting diminished GZMB-dependent activation of apoptotic pathways in target cells (Fig. EV1D). In line with this finding, secretion of GZMB and PRF1 by CTL was markedly diminished (Fig. 1G). Unexpectedly, T2 CTL exhibited elevated intracellular levels of GZMB and PRF1 (Fig. 1H/I). This led us to speculate that the

transfer of cytotoxic granules (CG) to the immunological synapse and their subsequent exocytosis may be impaired, causing the retention of GZMB and PRF1 within the cell. Consistent with this, quantification of CG exocytosis, measured by plasma membrane incorporation of CD107a, showed a significant reduction in CD107a⁺ CTL at T2 (Fig. 1J). Taken together, these results demonstrate that, after major surgery, CTL display severe immunosuppression (^PITC^CTL), which may contribute to post-operative immune-related morbidity and mortality.

## Multi-omics approach reveals severe oxidative stress signature in patients after major surgery

To elucidate potential mechanisms underlying the observed ^PITC^CTL, we performed a comprehensive analysis of patient sera using proteomics and metabolomics. Compared to T1, serum proteomes at T2 and on the first postoperative day (T3) were profoundly altered with >100 proteins exhibiting significantly differential expression, which formed clearly distinguishable clusters (Fig. EV2A). As expected, several acute-phase proteins (CRP, LTA4H, LBP, and HSPA1B) were strongly induced, indicating a systemic inflammatory state (Fig. EV2B). Moreover, we detected significantly higher expression of numerous proteins related to redox homeostasis (TALDO1, PRDX1/2/6, SELENBP1, BLVRB, and AHCY; Fig. 2A) and to the oxidative stress response (PARK7, MPO, GSTO1, GSS, GCLC, CAT, and ADH1B; Fig. 2A).

LC-MS/MS-based untargeted metabolomic analyses (TOF-MS) of human sera revealed profound changes to the metabolic fingerprint following major surgery (Figs. 2B–D and EV2C). Metabolite quantification showed reduced levels of protective metabolites, such as Trigonelline, Aminobenzamide, Bilirubin, and Citrulline, as well as metabolites involved in DNA repair (Fig. 2E). In contrast, metabolites associated with ROS production and lipid peroxidation, including Decanoylcarnitine and oxidized N-acetylcysteine (Diacetylcysteine), were significantly increased post-surgery (Fig. 2E). Functional analysis highlighted the enrichment of pathways related to redox processes and mitochondrial functions (Fig. 2F,G). Joint pathway integration of proteomic and metabolomic data corroborated changes to metabolism and redox homeostasis (Fig. 2H). These findings indicate that major surgery induces a distinct serum signature reflecting systemic oxidative stress at both metabolomic and proteomic levels.

## MDSC as the predominant source of reactive oxygen species (ROS) after major surgery

We next investigated whether the observed oxidative stress following PITC could be attributed to the activation of immune cells. To address this, we quantified transcript levels of the two key ROS-producing enzymes, NADPH oxidase 2 (NOX2) and myeloperoxidase (MPO) (Winterbourn et al, 2016), in freshly isolated lymphocytes, monocytes, and neutrophils. Neither CD4⁺ nor CD8⁺ T-cells or monocytes showed upregulation of ROS-producing enzymes (Fig. 3A,B). Even neutrophils, typically known for their ROS-producing capacity, exhibited only a slight increase in NOX2 and MPO mRNA. However, a marked upregulation of both enzymes was observed in the overall PBMC population, suggesting that a previously unrecognized cell entity emerging acutely after major surgery may be responsible for the systemic oxidative stress response.

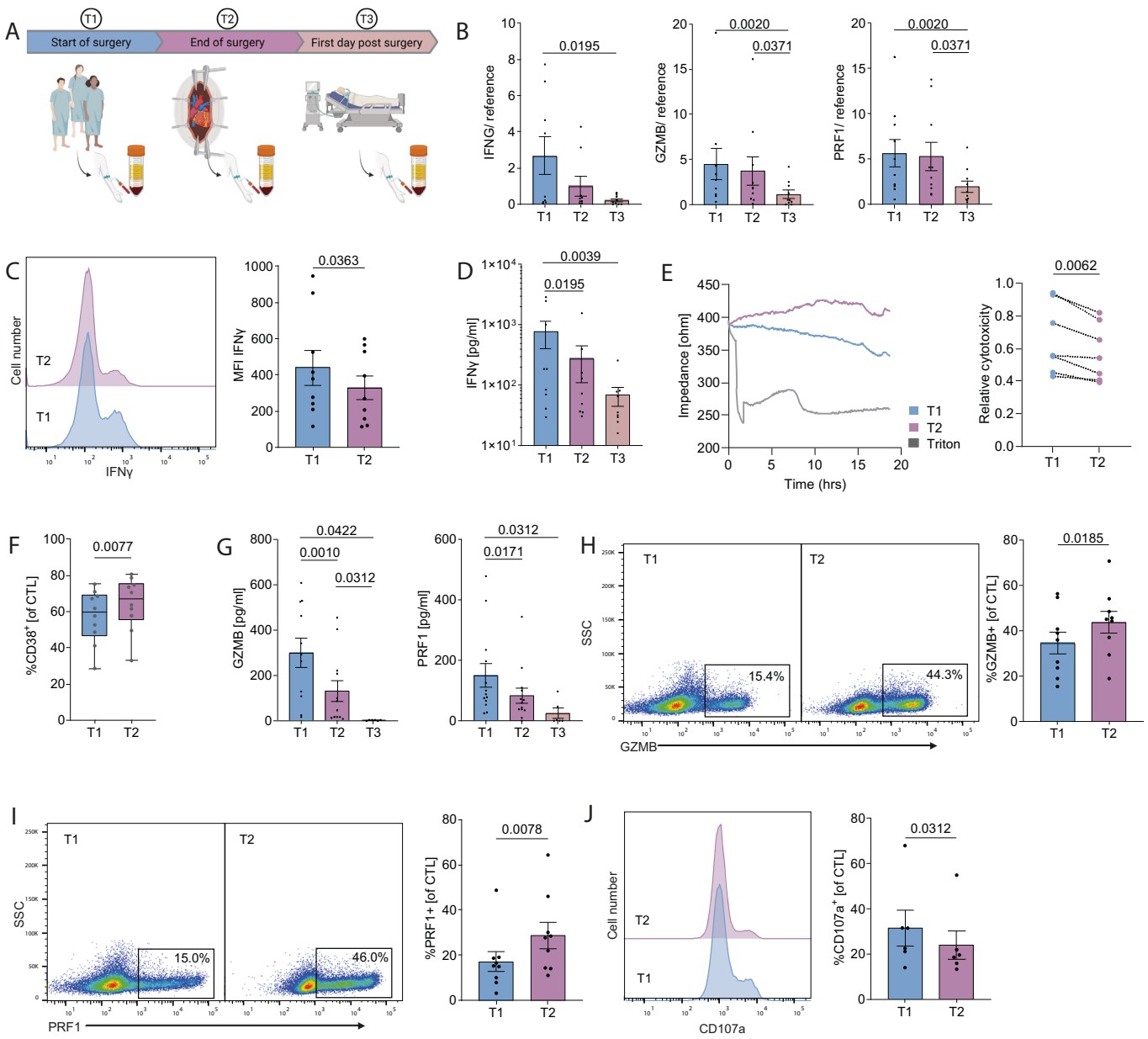

**Figure 1. Post surgical immunosuppression (PITC) of CTL.**

(A) Scheme of the study flow and time points of blood withdrawal during the course of major surgery. (B) mRNA expression of *IFNG* (left), *GZMB* (middle) and *PRF1* (right) in CTL (+CD3/CD28 dynabeads, stimulated for 18 h) relative to endogenous controls, n = 9/10/10 individual patients. (C) Flow cytometric analysis (FCA) of intracellular IFN-γ staining and quantification in CTL (+CD3/CD28 dynabeads, stimulated for 18 h; re-stimulation with PMA/Ionomycin for 4 h), n = 9 individual patients. (D) Quantification of IFN-γ secretion by CTL (+CD3/CD28 dynabeads, stimulated for 18 h), n = 9 individual patients. (E) Representative impedance plot (left) and quantification of relative CTL cytotoxicity (right) using ECIS, n = 7 individual patients. (F) FCA of CD38 staining and quantification in CTL, minimum = 28.6/32.9, 25%-percentile = 46.73/55.65, median = 59.65/67.05, 75%-percentile = 69.10/75.75, maximum = 75.10/80.60 (T1/T2), n = 10 individual patients. (G) Quantification of GZMB (left) and PRF1 (right) secretion by CTL (+CD3/CD28 dynabeads, stimulated for 18 h) as measured by ELISA, n = 11/12/6 (GZMB) and 13/13/6 (PRF1) individual patients. (H) FCA of intracellular GZMB staining and quantification in CTL (+CD3/CD28 dynabeads, stimulated for 18 h; re-stimulation with PMA/Ionomycin for 4 h), n = 9 individual patients. (I) FCA of intracellular PRF1 staining and quantification in CTL, n = 9 individual patients. (J) FCA of CD107a staining and quantification in CTL (+CD3/CD28 dynabeads, stimulated for 18 h; re-stimulation with PMA/Ionomycin for 4 h), n = 6 individual patients. If not stated otherwise, data are represented as mean ± SEM. P values as indicated, paired *t*-test or Wilcoxon matched-pairs signed-rank test, as appropriate. Source data are available online for this figure.

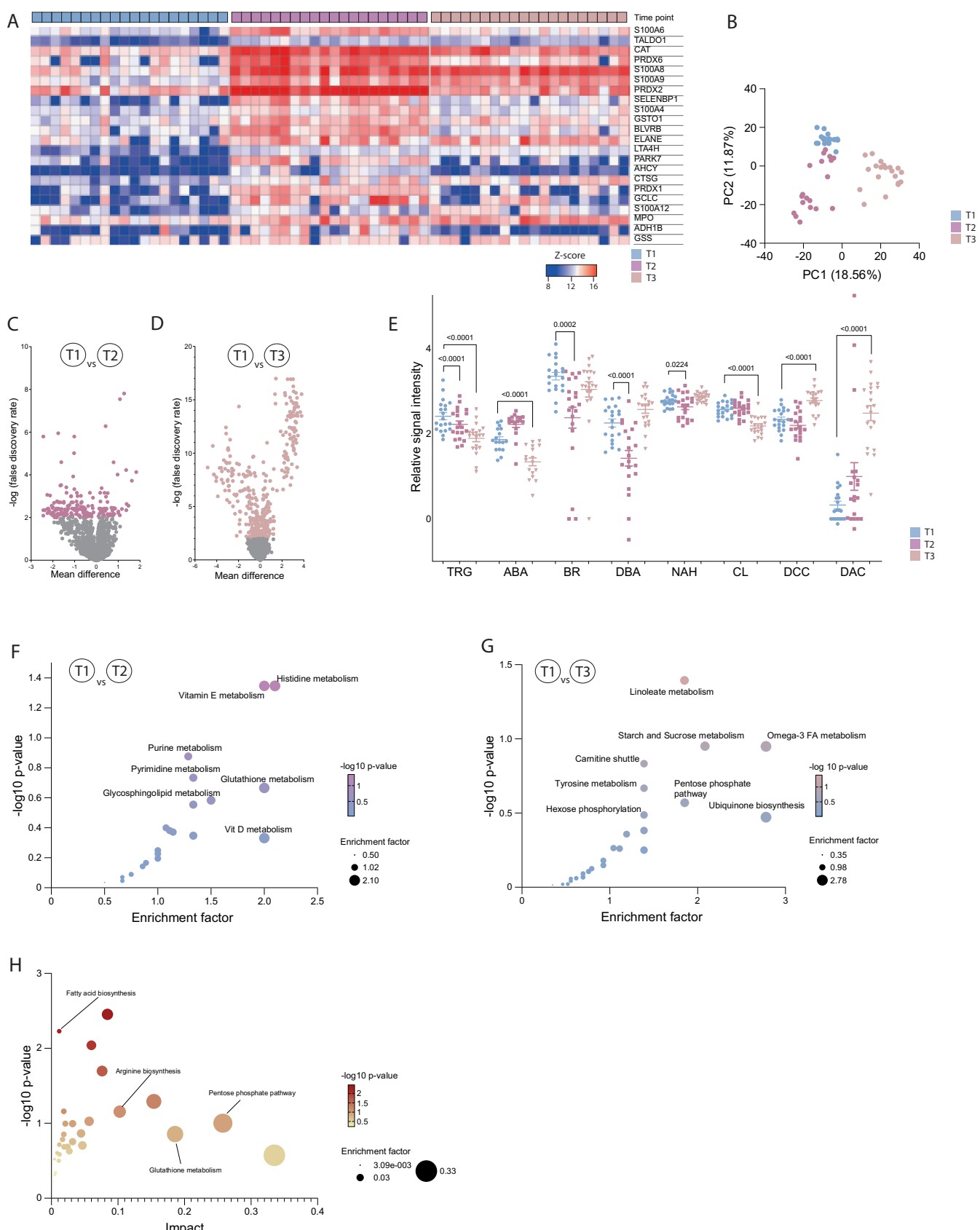

**Figure 2.   Serum proteomics and metabolomics reveal an oxidative stress signature after major surgery.**

(A) Heat map depicting quantity of significantly differentially expressed serum proteins associated with ROS production or redox homeostasis. Z-scores and time points as indicated, $n = 20$ individual patients. (B) Principal component analysis depicting metabolic profiling results during the course of major surgery, time points as indicated. (C) Volcano plot for metabolite abundance of comparisons T1 and T2. Metabolites with −log FDR >2 are colored. (D) Volcano plot for metabolite abundance of comparisons T1 and T3. Metabolites with −log FDR >2 are colored. (E) Mass spectrometric non-targeted serum metabolome analyses during the course of major surgery, time points as indicated. TRG trigenolline, ABA aminobezamide, BR bilirubin, DBA dihydroxybenzoic acid, NAH N-acetylhistidine, CL citrulline, DCC decanoylcarnitine, DAC diacetylcysteine. Relative quantification (normalized metabolite intensity, log 10, arbitrary units). Mean ± SEM with dots representing individual values, $n = 20$ individual patients, p values as indicated, paired t-test. (F) Functional analysis of untargeted metabolomics, displaying pathway enrichment results of results T1 compared to T2. (G) Functional analysis of untargeted metabolomics, displaying pathway enrichment results of results T1 compared to T3. (H) Joint pathways analysis of metabolomic and proteomic data. Source data are available online for this figure.

In a prior study, we demonstrated that cardiac surgery with cardiopulmonary bypass leads to the acute appearance of ROS-producing myeloid-derived suppressor cells (MDSC), which negatively impact immune responses (Hübner et al, 2019). We thus hypothesized that MDSC induction might be a general response to major surgery and, accordingly, investigated their presence in our cohort of patients. As shown in Fig. 3C, major surgery also led to the emergence of CD14⁻HLA-DR⁻CD11b⁺CD15⁺ MDSC (Figs. 3C and Fig. EV3A–C), whereas minor surgeries, such as joint arthroscopic procedures, did not induce MDSC formation (Fig. EV3D,E). These findings suggest that the transient appearance of MDSC may represent a common response to major surgery.

To further characterize MDSC as the potential cause of oxidative stress after major surgery, we conducted a comprehensive analysis of freshly isolated T2 MDSC using Next-Generation Sequencing. Differential gene expression analysis revealed a unique transcriptional profile in MDSC, with over 6000 downregulated and 8000 upregulated genes compared to neutrophils, their closest relatives (Fig EV3F–H) (Veglia et al, 2021). Many of these upregulated genes are involved in ROS generation, redox homeostasis, and the antioxidative stress response (Fig. 3D). Notably, NOX2 and MPO were strongly expressed in MDSC. These findings were further corroborated by qRT-PCR-based transcript quantification, which showed nearly exclusive expression of NOX2 and MPO mRNA in MDSC, with transcript levels being up to nearly 50 times higher than in neutrophils (Fig. 3E,F).

A hallmark immunosuppressive mechanism of MDSC is their rapid release of ROS or ROS-producing proteins such as MPO and NOX2 into the extracellular milieu, rather than intracellular ROS storage in granules that is mainly seen in neutrophils (Ohl and Tenbrock, 2018; Huang et al, 2023). As expected, T2 MDSC exhibited no evidence of elevated intracellular ROS levels (Fig. EV3H) as measured by CellROX-dye. In contrast, luminol-based chemiluminescence assays directly quantify extracellular ROS production. Importantly, sorted T2 MDSC displayed markedly higher luminol-induced relative luminescence compared to T2 neutrophils, indicating substantially greater extracellular ROS release and/or peroxide-dependent enzyme activity of MDSC (Fig. 3G). Together, these data identify MDSC as the dominant cellular source of surgery-induced oxidative stress, acting primarily through extracellular ROS generation.

## ROS-release by MDSC compromises CTL mitochondrial function

To investigate whether oxidative stress evoked by MDSC impairs CTL function and thus contributes to postoperative immune

paralysis, we quantified intracellular ROS in ᴾᴵᵀᶜCTL, revealing markedly elevated levels after major surgery. (Fig. 4A). Concurrently, the antioxidative capacity, as reflected by intracellular reduced glutathione (GSH) levels, was significantly diminished (Fig. 4B).

Critical and prolonged elevation of ROS in CTL could be the result of oxidative damage to their mitochondria, resulting in dysfunctional ROS generation (Martínez-Reyes and Cuezva, 2014). To further explore this, we assessed mitochondrial membrane potential (ΔψM) as a marker of mitochondrial integrity. ᴾᴵᵀᶜCTL exhibited significantly impaired ΔψM, accompanied by a concomitant increase in mitochondrial ROS levels (Fig. 4C,D), indeed suggesting a disruption of mitochondrial coupling (Guo et al, 2013; Peoples et al, 2019). ᴾᴵᵀᶜCTL further exhibited a significant decrease in Δψ-independent MitoTracker Green labeling and a marked reduction in the expression of respiratory chain (OXPHOS) protein complexes, indicating a lower mitochondrial protein content (Fig. 4E,F). Of note, we detected a preferential decrease of complex I/IV and complex V (Fig. 4G), which is known to be associated with detrimental ROS production in conditions of mitochondrial stress (Murphy, 2009; Leadsham et al, 2013).

These mitochondrial alterations were associated with a substantial decline in mitochondrial function, reflected by a strong reduction in all parameters of respiratory capacity (Fig. 4H). Basal and maximal oxidative respiration in ᴾᴵᵀᶜCTL were significantly reduced (Fig. 4I,J), while spare respiratory capacity and ATP-linked respiration were also diminished (Fig. 4K,L). Notably, compromised oxidative respiration was not compensated by an increase in glycolytic capacity (Fig. EV4A). In line with this, cellular glucose uptake was significantly reduced in ᴾᴵᵀᶜCTL (Fig. 4M). To further confirm that MDSC-derived ROS are responsible for the observed mitochondrial dysfunction in T2 CTL, we co-cultured preoperative T1 CTL with either T1/T2 neutrophils or MDSC. Upon co-culture, T1 CTL exhibited a phenotype highly similar to T2 CTL only when co-cultivated with MDSC, characterized by impaired mitochondrial mass and decreased mitochondrial membrane integrity (Fig. 4N,O) as well as a marked increase in (m)ROS (Fig. 4P,Q). In contrast, T2/T1 neutrophils did not significantly impact CTL mitochondrial health (Fig. 4N–Q).

ROS-producing enzymes like MPO might also be secreted by MDSC, leading to ROS formation in the serum (Monteseirín et al, 2001). To assess the impact of such released enzymes and other serum factors on CTL, we next incubated T1 CTL with T2 serum, which again induced a CTL phenotype of mitochondrial dysfunction characterized by markedly distorted mitochondrial mass (Fig. EV4B), compromised mitochondrial membrane potential (Fig. EV4C) and impaired mitochondrial respiratory function (Fig. EV4D–G).

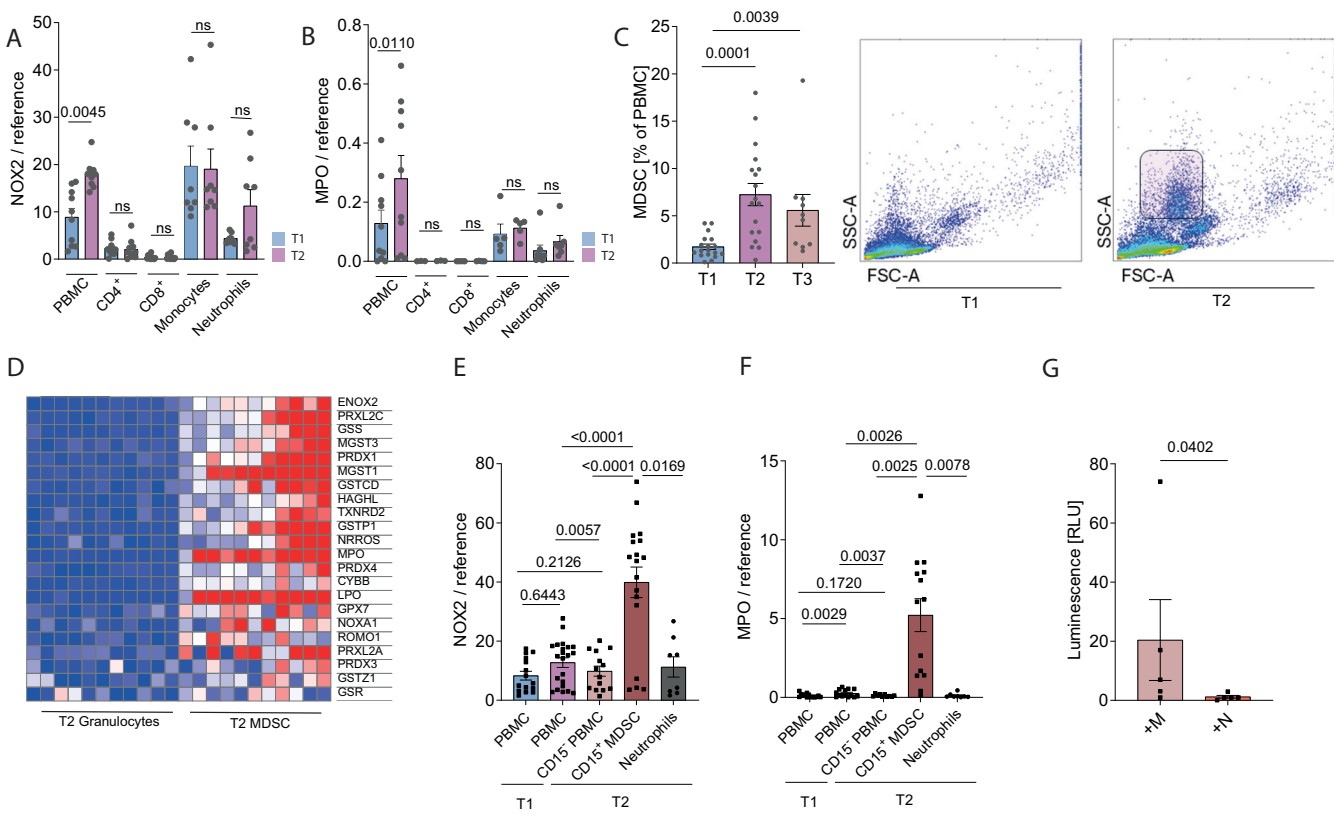

**Figure 3. ROS burst after major surgery by the transient occurrence of myeloid-derived suppressor cells.**

(A, B) mRNA expression of *NOX2* (A) and *MPO* (B) in PBMC, CD4+ T-cells, CTL, monocytes and neutrophils relative to endogenous control. Time points as indicated, $n = 10/10/11/8/8$ (NOX2) and $n = 10/3/4/5/7$ (MPO) individual patients. (C) Quantification of CD14-HLA-DR-CD11b+CD15+ MDSC and representative FCA before (T1) and after (T2) major surgery, $n = 18/18/10$. (D) Heat map showing quantity of significantly differentially expressed genes associated with oxidative stress of T2 MDSC compared to T2 granulocytes. Genes as indicated. Red color indicates an upregulation, and downregulated genes are indicated by blue color. (E, F) mRNA expression of *NOX2* (E) and *MPO* (F) in PBMC, CD15- PBMC, CD15+ MDSC and granulocytes, relative to endogenous control. Time points as indicated. $n = 15/22/14/18/8$ (NOX2) and $n = 14/14/10/14/8$ (MPO) individual patients. (G) Luminol-based assay and quantification of extracellular ROS ($H_2O_2$- and peroxidase-dependent ROS production) of freshly sorted T2 MDSC (+M) compared to T2 neutrophils (+N), $n = 5$ individual patients. Data were represented as mean ± SEM with dots representing individual values. *P* values as indicated, paired *t*-test, Wilcoxon matched-pairs signed-rank test or log normal Welch's *t*-test (G), as appropriate. Source data are available online for this figure.

## Stress-induced mitochondrial hyperfusion in response to major surgery impairs mitochondrial translocation to the immunological synapse

We next analyzed isolated mitochondria from PITCCTL at time points T1 and T2, based on our previous experiments that indicate the most profound mitochondrial damage immediately after major surgery. This analysis again revealed a significant reduction in membrane potential, protein content, and OXPHOS protein complexes (Fig. 5A–C). These findings suggest that the impaired mitochondrial fitness of PITCCTL is not solely due to a reduction in the overall mitochondrial content but also reflects dysfunction at the level of individual organelles. To preserve basal metabolic function following stress-induced damage, mitochondria are known to undergo fusion processes (Tondera et al, 2009). In isolated mitochondria of PITCCTL, we observed a significant higher protein expression of the fusion proteins Mitofusin 2 (MFN2) and Optic atrophy 1 (OPA1), while the expression of its functional counterpart, fission protein DRP1, was downregulated (Fig. 5D/E). These expression alterations suggest a shift in the balance between mitochondrial fusion and fission, favoring fusion in PITCCTL. Consistent with this, confocal microscopy

revealed distinct structural changes, with mitochondria forming large supercomplexes, resulting in a reduced mitochondrial count per cell and an increase in mitochondrial length and mean mitochondrial area (Fig. 5F–H). DRP1-dependent mitochondrial fission enables mitochondrial translocation to the T-cell immunological synapse (IS), ensuring ATP production in close proximity to the T-cell receptor (TCR) (Lisci and Griffiths, 2023; Baixauli et al, 2011). To examine this in PITCCTL, we assessed the subcellular localization of the TCR and mitochondria. As shown in Fig. 5I, we observed a significant reduction in mitochondria-TCR approximation. These findings indicate that PITC-induced oxidative stress triggers mitochondrial hyperfusion in CTL, impairing their ability to translocate to the IS.

## Mitochondrial dysfunction contributes to PITC

We next aimed to determine whether isolated mitochondrial damage in intact CTL produces the same functional phenotype observed in PITCCTL, thereby providing further evidence that mitochondrial dysfunction underlies the decrease of immunometabolic fitness in PITCCTL. To investigate this, we activated human T cells from healthy donors and treated them with the mitochondria-

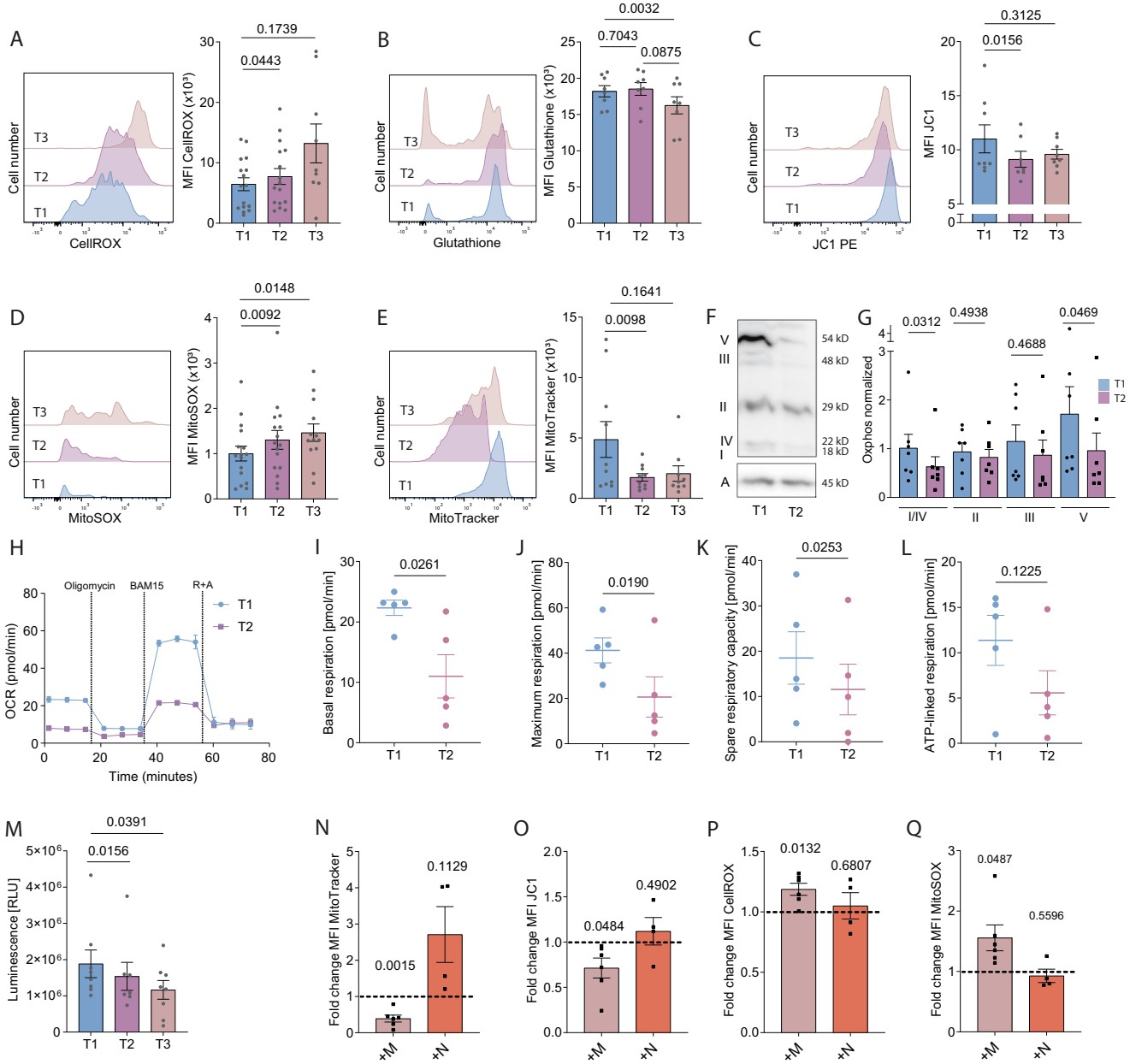

**Figure 4. Reactive oxygen species overpower CTL antioxidative capacity and leads to cellular metabolic dysfunction.**

(A) FCA of CellROX green staining and quantification, $n = 16/16/9$ individual patients. (B) FCA of ThioTracker (GSH content) staining and quantification, $n = 8$ individual patients. (C) FCA of JC1 staining and quantification of red/green fluorescence ratio, $n = 8$ individual patients. (D) FCA of MitoSOX staining and quantification, $n = 16/16/13$ individual patients. (E) FCA of MitoTracker green staining and quantification, $n = 10/10/9$ individual patients. (F) Exemplary immunoblot of OXPHOS complexes in CTL. I–V = complex I–V; A beta-actin. Subunits detected by the antibody cocktail: complex I: subunit NDUFB8; complex II: subunit 30 kDa (SDHB); complex III: subunit Core 2 (UQCRC2); complex IV: subunit II (COXII); complex V: ATP synthase subunit alpha (ATP5). (G) Normalized densitometric quantification of individual ETC complexes as indicated, $n = 7$ individual patients. (H) Representative seahorse OCR plot. (I–L) Seahorse quantification of (I) basal OCR, (J) maximal OCR, (K) spare respiratory capacity (SRC), and (L) ATP production in CTL, $n = 5$ individual patients. (M) Glucose uptake measured by fluorescence quantification, $n = 8/7/8$ individual patients. (N–Q) Co-cultivation of T1 CTL with MDSC (+M) or with T2 neutrophils (+N), depicted as fold change compared to T1 CTL (+M) or T1 CTL with T1 neutrophils (+N). (N) FCA of MitoTracker green staining and quantification, $n = 6/4$ individual patients. (O) FCA of JC1 staining and quantification of red/green fluorescence ratio, $n = 6/4$ individual patients. (P) FCA of CellROX green staining and quantification $n = 6/4$ individual patients. (Q) FCA of MitoSOX red staining and quantification $n = 6/4$ individual patients. Data were represented as mean ± SEM with dots representing individual values. P values as indicated, paired t-test, Wilcoxon matched-pairs signed-rank test and one-sample t-test (N–Q), as appropriate. Source data are available online for this figure.

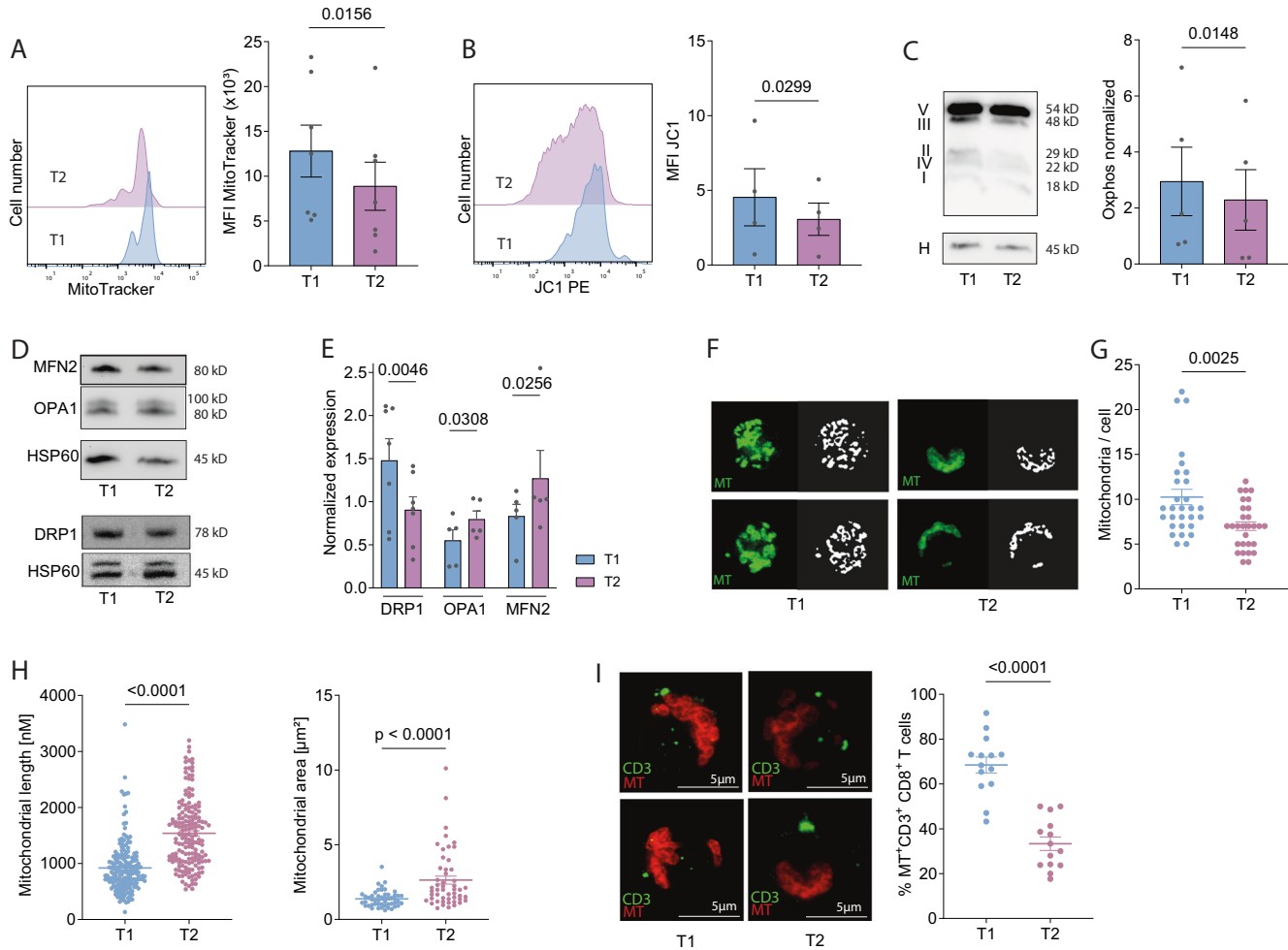

**Figure 5. PITC leads to stress-induced mitochondrial hyperfusion in CD8$^+$ CTL.**

(A) FCA of MitoTracker green and quantification in isolated mitochondria from CTL, $n = 7$ individual patients. (B) FCA of JC1 green and quantification of red/green fluorescence in isolated mitochondria from CTL, $n = 4$ individual patients. (C) Immunoblot of OXPHOS complexes using isolated mitochondria of CTL and normalized densitometric quantification of complex V, $n = 5$ individual patients. I–V complex I-V; H heat shock protein 60 (HSP60). Subunits detected by the antibody cocktail: complex I: subunit NDUFB8; complex II: subunit 30 kDa (SDHB); complex III: subunit Core 2 (UQCRC2); complex IV: subunit II (COXII); complex V: ATP synthase subunit alpha (ATP5). (D, E) Representative immunoblot of mitochondrial fusion and fission proteins of isolated mitochondria of CTL (D) and normalized densitometric quantification (E) of the respective proteins, $n = 5/4/3$ individual patients. (F) Representative projections of 3D image stacks of CTL stained with MitoTracker (MT, green, left) and the respective skeleton images (white, right), generated using mitochondrial analyzer, time points as indicated, representative of six individual patients. (G) Quantification of mitochondrial count per cell, $n = 29$ cells from three individual patients. (H) Quantification of mitochondrial length (left) and mean mitochondrial area using mitochondrial analyzer (right), $n = 180/170$ mitochondria from 14/22 cells from three individual patients (length) and $n = 50$ cells from three individual patients (area). (I) Representative projection of 3D image stacks (two per time point) of translocated mitochondria to the IS in CTL as indicated by MitoTracker (MT) Deep Red and CD3 FITC staining, and quantification of mitochondrial translocation to the CTL IS, time points as indicated, $n = 14$ from 5 individual patients. If not stated otherwise, data were represented as mean ± SEM. $P$ values as indicated, paired $t$-test or Wilcoxon matched-pairs signed-rank test (A-E), and unpaired $t$-test or Mann–Whitney test (G–I), as appropriate. Source data are available online for this figure.

specific protonophore FCCP, which partially uncouples the respiratory chain and mildly reduces $\Delta\psi M$ ($\Delta\psi M^{low}$CTL). A concentration of 20 nM FCCP - ~1/1000 of the typical dose required for complete uncoupling - resulted in a reduction of mitochondrial membrane potential comparable to that observed in $^{PITC}$CTL, without compromising cell viability (Figs. 6A and EV5A). In $\Delta\psi M^{low}$ CTL, we observed diminished mitochondrial function, marked by impaired oxidative phosphorylation and an absence of compensatory glycolysis, resembling the metabolic changes seen in $^{PITC}$CTL (Fig. 6B–D). Furthermore, mitochondrial protein content

was significantly reduced (Fig. 6E) and a shift toward mitochondrial fusion was evident, as indicated by the downregulation of DRP1 and upregulation of MFN2 expression (Fig. 6F,G). This mitochondrial dysfunction in $\Delta\psi M^{low}$ CTL led to a functional phenotype similar to that of $^{PITC}$CTL, characterized by reduced T-cell proliferation and significantly lower mRNA expression and protein secretion of IFNG/IFN-γ, GZMB, and PRF1 (Fig. 6H–J). Collectively, these findings strongly support the notion that mitochondrial damage drives the decrease of immunometabolic fitness in $^{PITC}$CTL.

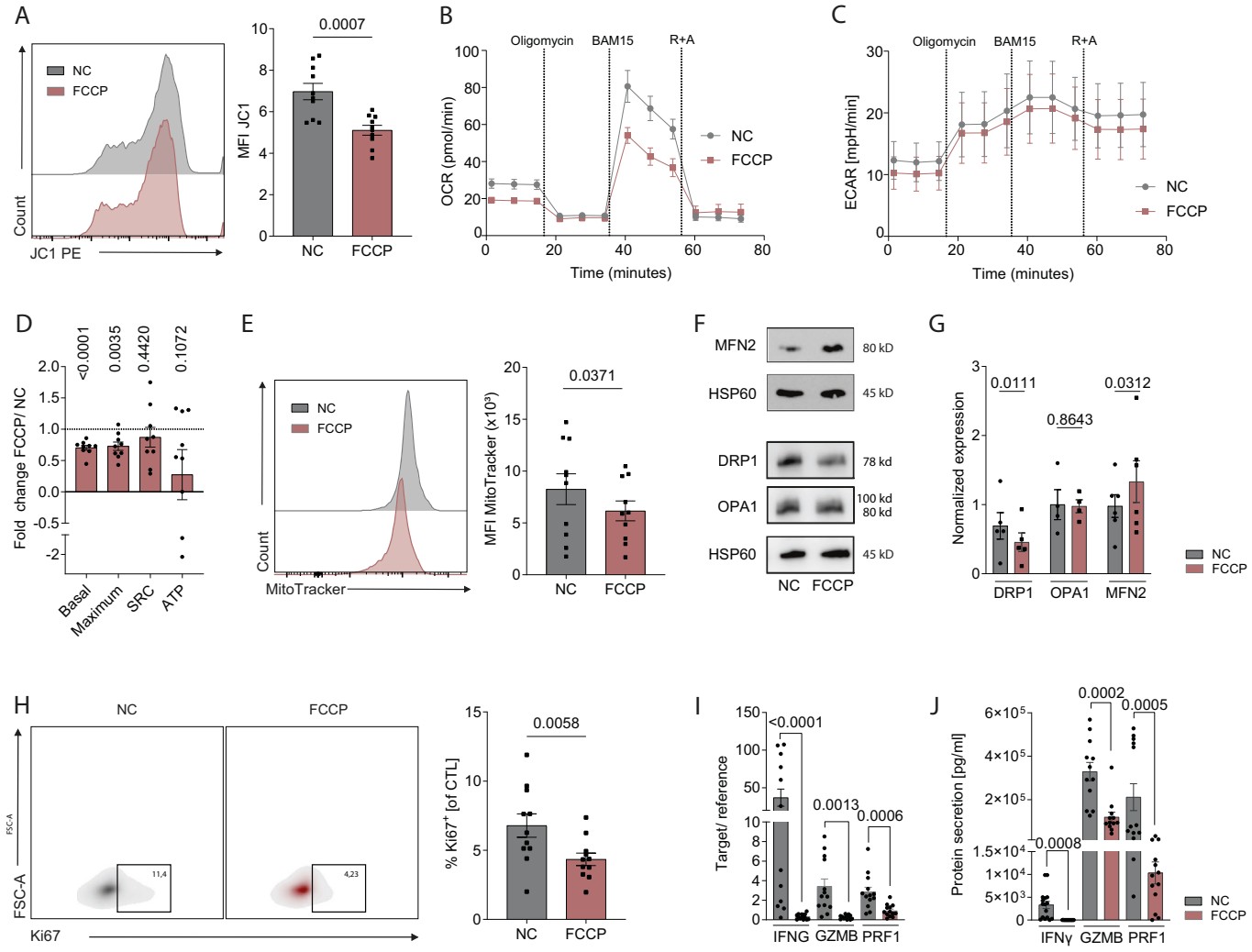

**Figure 6. Mitochondrial membrane potential is required for T-cell immunocompetence.**

(A) FCA of JC1 and quantification of red/green fluorescence in CTL, $n = 10$ individual biological replicates. (B) Representative Seahorse OCR plot. (C) Pooled seahorse ECAR plot, mean ± SEM from ten individual biological replicates. (D) Fold change of basal OCR, maximum OCR, spare respiratory capacity (SRC) and ATP production, $n = 10$ individual biological replicates. (E) FCA of MitoTracker green and quantification in CTL, $n = 10$ individual biological replicates. (F) Immunoblot of mitochondrial fusion and fission proteins of isolated mitochondria of CTL and (G) normalized densitometric quantification of the respective proteins, $n = 5/4/6$ individual biological replicates. (H) FCA of Ki-67 and quantification in CTL, $n = 11$ individual biological replicates. (I) mRNA expression of *IFNG*, *GZMB* and *PRF1* of CTL, relative to endogenous controls, $n = 14/13/13$ individual biological replicates. (J) Quantification of IFN-γ/ GZMB/ PRF1 secretion by CTL, $n = 16/12/13$ individual biological replicates. If not stated otherwise, data were represented as mean ± SEM. *P* values as indicated, one-sample *t*-test (D), paired *t*-test or Wilcoxon matched-pairs signed-rank test, as appropriate. Source data are available online for this figure.

## Enhancing mitochondrial antioxidative capacity mitigates PITC

Finally, we aimed to investigate whether mitochondria-specific interception of ROS could serve as a potential strategy to mitigate PITC. To this end, PITCCTL were supplemented with the mitochondria-targeted antioxidant MitoTEMPO (MT). As shown in Fig. 7A–C, PITCCTL treated with MT exhibited a significant increase in mitochondrial protein content (Fig. 7A) and a marked improvement in mitochondrial function, characterized by enhanced oxidative phosphorylation and preserved glycolytic capacity (Figs. 7B and EV5B). Both basal and maximal respiration, as well as spare respiratory capacity, were significantly elevated (Fig. 7C). MT

treatment also alleviated stress-induced mitochondrial hyperfusion, as evidenced by a significant reduction in mean mitochondrial length and an increase in mitochondrial count per cell (Fig. 7D,E). Additionally, a substantial increase in mitochondrial translocation to the T-cell immunological synapse was detected (Fig. 7F,G). Functionally, MT treatment restored the effector functions of PITCCTL, as demonstrated by ECIS cytotoxicity assays, which revealed a significant improvement of cytolytic capacity (Fig. 7H,I). Of note, these effects were only present under conditions of critically elevated oxidative stress, as T1 CTL supplemented with MT showed no alterations to mitochondrial mass, mitochondrial function or CTL cytotoxicity (Figs. 7B and EV5C–E). By contrast, T1 CTL that have been damaged via co-cultivation with T2 MDSC exhibit improved mitochondrial

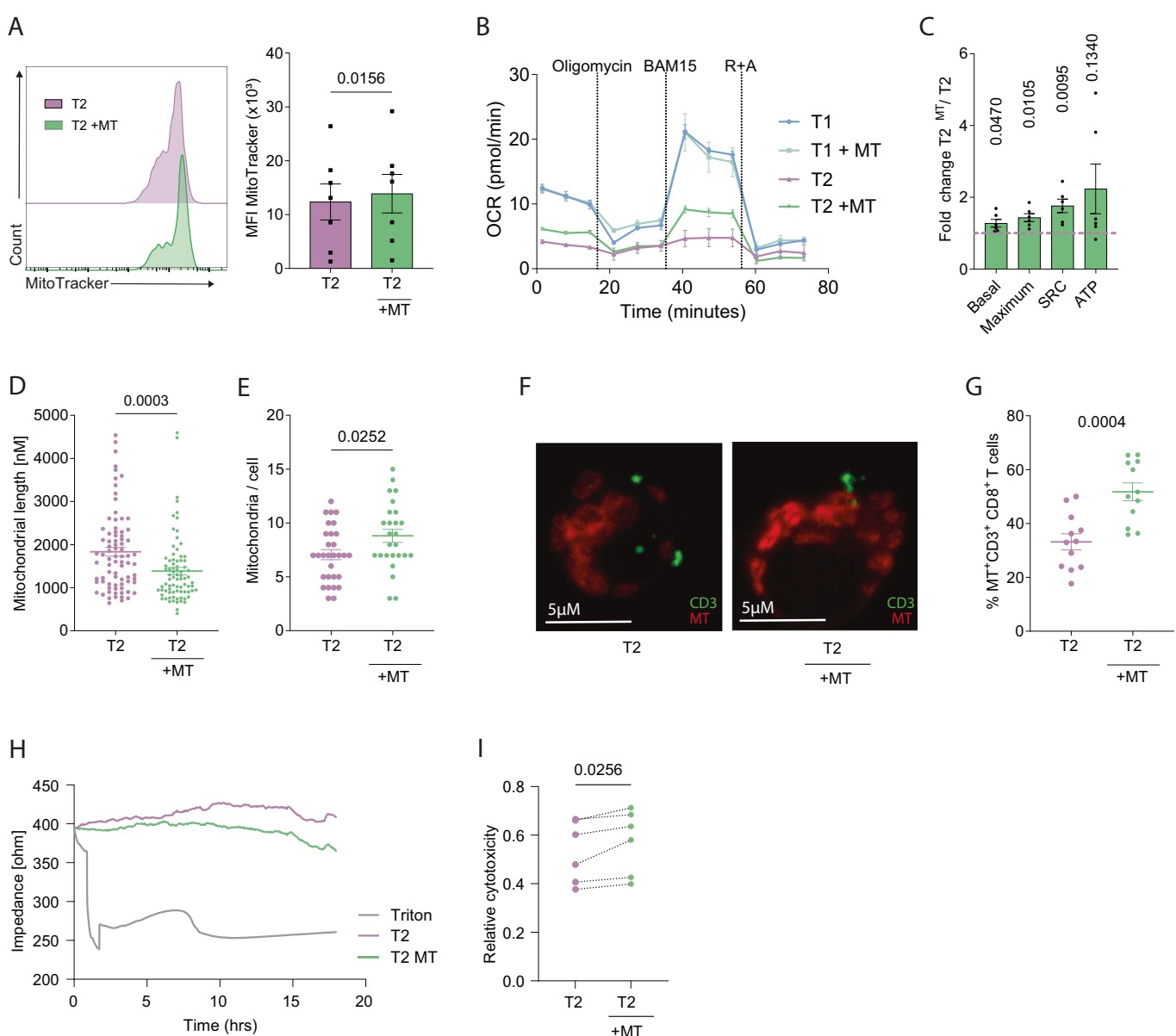

**Figure 7. Enhancing mitochondrial antioxidative capacity using MitoTEMPO (MT) ameliorates PITC.**

(A) FCA of MitoTracker green and quantification in CD8+ CTL, $n = 7$ individual patients. (B) Representative Seahorse OCR plot. (C) Fold change of basal OCR, maximum OCR, spare respiratory capacity (SRC) and ATP production, $n = 6$ individual patients. (D) Quantification of mitochondrial length, $n = 92$ mitochondria from 28 cells from three individual patients. (E) Quantification of mitochondrial count per cell, $n = 30/26$ cells from three individual patients. (F) Representative confocal microscopy image of mitochondria translocated to the TCR in CTL as indicated by MitoTracker (MT) Deep Red and CD3 FITC staining, T2 image as depicted in Fig. 5I, upper right image. (G) Quantification of mitochondrial translocation in proximity to the CTL IS, $n = 12$ from four individual patients. (H) Representative impedance plot and (I) quantification of relative CTL cytotoxicity using ECIS, $n = 6$ individual patients. If not stated otherwise, data are represented as mean ± SEM. P values as indicated, one-sample $t$-test (C), paired $t$-test or Wilcoxon matched-pairs signed-rank test (A–C, H, I), and unpaired $t$-test or Mann–Whitney test (D–G), as appropriate. Source data are available online for this figure.

health in response to MT treatment (Fig. EV5F). In conclusion, these findings both consolidate the detrimental role of oxidative stress for CTL immunometabolism and highlight potential therapeutic avenues.

## Discussion

Dysfunction of cytotoxic T cells following major surgery (PITCCTL) is recognized as a hallmark of postoperative immune paralysis

(Albertsmeier et al, 2015; Sun et al, 2017; Ward et al, 2008), with significant clinical consequences, including increased rates of nosocomial infections, prolonged ICU stays, and elevated perioperative mortality (Hadaya et al, 2022; Torrance et al, 2016; Pérez-Granda et al, 2024; Hadaya et al, 2021). The underlying mechanisms of this phenomenon remain poorly understood. In the present study, severe systemic oxidative stress, produced by transiently emerging myeloid-derived suppressor cells (MDSC) was identified as a key driver of this dysfunction. MDSC-derived

reactive oxygen species (ROS) caused a critical increase in ROS levels within CTL, overwhelming their antioxidative defenses and resulting in mitochondrial damage, ultimately leading to decreased metabolic fitness of CTL.

In this study, we selected patients scheduled for major surgery who did not have preexisting conditions associated with immune dysregulation, such as cancer, autoimmune diseases, immunosuppression, or active infections. We demonstrated that postoperative CTL of these patients indeed exhibited clear signs of immunoparalysis, as evidenced by decreased expression of IFNG, PRF1, and GZMB, and impaired cytotoxicity. To evaluate the root cause of this disorder, we employed a multi-omics approach that identified a systemic serum signature indicative of severe oxidative stress. We identified acutely emerging MDSC as the primary source of ROS after major surgery. Other immune cell types contributed minimally or not at all to ROS production in this scenario. Lymphocytes and monocytes generally produce low levels of ROS, while neutrophils are known to generate ROS-bursts as part of their antimicrobial defense against pathogens, rather than during sterile inflammation (Dale et al, 2008; Taylor et al, 2013; El-Benna et al, 2016; Botha et al, 1995; Lumsdaine et al, 2014). Consistently, we did not observe activation of the key ROS-producing enzymes in these leukocyte populations and found markedly higher ROS levels produced by MDSC than by pre- or postoperative neutrophils.

MDSC are commonly observed in chronic inflammatory conditions such as cancer and autoimmunity, where they are thought to suppress immune responses through the production of ROS and other immunosuppressive molecules (Beury et al, 2016; Groth et al, 2019). In the perioperative setting, only one study has previously reported the presence of MDSC following the use of cardiopulmonary bypass (Hübner et al, 2019). However, the current study expands on this by demonstrating that MDSC emerge after major surgery, regardless of the specific surgical procedure. In contrast, minor surgeries do not lead to a significant increase in MDSC numbers. The origins and precise mechanisms underlying MDSC release remain speculative. One may believe that these cells are released from the bone marrow, potentially serving an evolutionary function in response to life-threatening injuries outside of a sterile environment. In such a context, MDSC may help maintain a balance in the inflammatory response, preventing excessive immune activation that could be detrimental.

However, in the sterile environment of major surgery, excessive ROS production by MDSC may impair CTL function, thereby contributing to the postoperative immunosuppression observed. Our findings show a significant increase in ROS levels in PITCCTL, while ROS-producing enzymes were not induced. Additionally, we observed a depletion of reduced glutathione (GSH), a critical antioxidant, indicating the exhaustion of the cellular compensatory mechanisms.

In CTL, ROS levels must be tightly regulated. At low concentrations, ROS function as essential signaling molecules for immune cell activation (Picard and Shirihai, 2022; Hirschberger et al, 2021). However, excessive ROS levels induce mitochondrial damage by inactivating multiple enzymes of the electron transport chain (ETC), thereby impairing oxidative phosphorylation (OXPHOS) (Guo et al, 2013; Vardhana et al, 2020). Furthermore, elevated ROS levels can damage the mitochondrial membrane, exacerbating OXPHOS dysfunction and triggering the release of

ROS from the mitochondrial matrix, which perpetuates a vicious cycle of mitochondrial damage (Zorov et al, 2014). In this study, we identified all the hallmark features of ROS-induced mitochondrial damage in PITCCTL: A reduction in mitochondrial mass, decreased expression of ETC proteins, impaired OXPHOS function, and compromised mitochondrial membrane integrity.

Furthermore, we observed signs of stress-induced mitochondrial hyperfusion (SIMH) in PITCCTL. Specifically, we detected elevated levels of fusion proteins, reduced mitochondrial expression of the fission protein Drp1, a decrease in the mitochondrial count per cell, and an increase in the mean mitochondrial length of PITCCTL. SIMH is a well-documented response to critical cellular damage, by which mitochondria merge partially damaged or dysfunctional organelles into larger supercomplexes (Das and Chakrabarti, 2020; Youle and van der Bliek, 2012; Buck et al, 2016). This process helps to maintain basal metabolic function and mitigate stress-induced damage; however, it occurs at the expense of mitochondrial mobility due to the downregulation of DRP1.

Drp1 has been shown to be essential for mitochondrial movement toward the T-cell immunological synapse, which is necessary to meet the high energy demands at this immune hub during T-cell activation (Lisci and Griffiths, 2023; Baixauli et al, 2011; Desdín-Micó et al, 2018; Quintana et al, 2007; Huppa and Davis, 2003). We found significantly reduced co-localization of mitochondria with the TCR in PITCCTL, indicating a substantial impairment in mitochondrial translocation. Consequently, PITCCTL not only show a massive decrease in mitochondrial oxidative phosphorylation but might also fail to localize their ATP synthesis to the potential area of highest demand. This bioenergetic failure likely contributes to impaired effector functions, including both the production and release of cytolytic granules (CG), as trafficking of these CG to the cell membrane is based on high ATP-consuming dynein-dependent microtubule–cytoskeleton interactions (Stinchcombe et al, 2006; de Saint Basile et al, 2010). Consistent with this, PITCCTL showed impaired release of CG.

The exact mechanism by which MDSC-derived ROS induce this mitochondrial damage in CTL after major surgery remains to be elucidated. However, co-culture of postoperative MDSC with healthy CTL and incubation with postoperative serum both induced a similar phenotype of mitochondrial damage, indicating that MDSC-derived ROS or ROS-producing enzymes initiate this process. This likely triggers a cell-intrinsic amplification of mitochondrial ROS, worsening damage and suppressing CTL function. Thus, we propose a two-step model of postoperative CTL suppression: (i) primary damage from MDSC-derived factors, and (ii) secondary ROS amplification within CTL due to mitochondrial dysfunction.

This hypothesis was further validated through ex vivo experiments, where low-dose application of the protonophore FCCP to CTL from healthy donors successfully reproduced the PITC phenotype, which was characterized by all previously described hallmark features, including collapse of the mitochondrial membrane potential, impairment of mitochondrial metabolism, enhanced mitochondrial fusion, and subsequent CTL effector dysfunction. In contrast, ex vivo treatment of PITCCTL with the mitochondria-specific antioxidant MitoTEMPO partially reversed all aspects of mitochondrial dysfunction and partially restored immune effector functions. Notably, MitoTEMPO treatment of T1 CTL co-cultivated with MDSC was also able to ameliorate

mitochondrial damage, thereby suggesting a potential avenue for future therapeutic intervention.

Collectively, this study offers the first detailed insights into the mechanisms driving CTL immunometabolic dysfunction following major surgery, laying the groundwork for the development of therapeutic and preventive strategies aimed at improving post-operative immune function and, consequently, enhancing clinical outcomes.

# Methods

### Reagents and tools table

| Reagent/resource | Reference or source | Identifier or catalog number |
| --- | --- | --- |
| **Experimental models** | | |
| n/a | n/a | n/a |
| **Recombinant DNA** | | |
| n/a | n/a | n/a |
| **Antibodies** | | |
| CD3 | SCBT | sc-20047 |
| AF488 anti-mouse | Invitrogen | A21042 |
| CD38 APC | Miltenyi | 130-113-991 |
| CD69 FITC | Miltenyi | 130-112-801 |
| CD223 APC | Biolegend | 369211 |
| CD366 FITC | Biolegend | 345021 |
| CD25 PE | Biolegend | 302605 |
| CD15 PE | Biolegend | 301905 |
| CD279 BV421 | Biolegend | 329920 |
| HLA-DR Pacific Blue | Biolegend | 307623 |
| CD14 PerCp | Biolegend | 325631 |
| CD11b FITC | Biolegend | 301329 |
| CD8PerCp | Biolegend | 344747 |
| CD8 BV421 | Biolegend | 344707 |
| CD107a PE | Biolegend | 328609 |
| GZMB BV421 | Biolegend | 396413 |
| Perforin 1 APC | Biolegend | 308111 |
| Interferon-g FITC | Biolegend | 502506 |
| CD366 FITC | Biolegend | 345021 |
| CD223 APC | Biolegend | 369211 |
| CD274 BV421 | Biolegend | 329920 |
| Total OXPHOS Human WB Antibody Cocktail | Abcam | ab110411 |
| MFN2 | Cell Signaling Technology | 67589 |
| HSP60 | Cell Signaling Technology | 4870 |
| DRP1 | Cell Signaling Technology | 5391 |
| Phospho-DRP1 | Cell Signaling Technology | 3455 |

| Reagent/resource | Reference or source | Identifier or catalog number |
| --- | --- | --- |
| OPA1 | Cell Signaling Technology | 11925 |
| b-Actin | Cell Signaling Technology | 5125 |
| **Oligonucleotides and other sequence-based reagents** | | |
| UPL Probe #26 (PD1) from UPL-Set Probes #1-90 | Roche | 04683633001 |
| UPL Probe #59 (LAG) | Roche | 04688562001 |
| UPL Probe #3 (TIM3) from UPL-Set Probes #1-90 | Roche | 04683633001 |
| UPL Probe #54 | Roche | 04688511001 |
| TBP RT-PCR Assay Cy3 | Bio-Rad | 10031231 |
| RPL13A RT-PCR Assay FAM | Bio-Rad | 12001950 |
| PCR primers | This study | Table EV1 |
| **Chemicals, enzymes and other reagents** | | |
| Human TruStain FcX Receptor Blocking Solution | Biolegend | 422302 |
| Mounting Medium | Ibidi | 50001 |
| Mitotracker Deep Red | Cell Signaling Technology | 8778 |
| Mitotracker Green | Cell Signaling Technology | 9074 |
| Fixation/Permeabilization Concentrate | Invitrogen | 00-5123-43 |
| Fixation/Permeabilization Diluent | Invitrogen | 00-5223-56 |
| Fixation/Permeabilization Buffer | Invitrogen | 00-8333-56 |
| Cell Activation Cocktail | Biolegend | 42330 |
| MitoSOX Red | Thermo Fisher | M36008 |
| CellROX Green | Thermo Fisher | C10492 |
| ThiolTracker | Thermo Fisher | T10095 |
| JC1 Mitochondrial Membrane Potential Assay Kit | Cayman Chemical | 701560 |
| LightCycler 480 Probesmaster | Roche | 04887301001 |
| Ficoll | Sigma | 10771 |
| CD8 Microbeads | Miltenyi | 130-045-201 |
| CD15 Microbeads | Miltenyi | 130-046-601 |
| StraightFrom whole blood CD15 Microbeads | Miltenyi | 130-091-058 |
| Mitochondria Isolation Kit, Human | Miltenyi | 130-094-833 |
| RPMI 1640 | Sigma | R0883-100ML |
| Seahorse XF RPMI Medium | Agilent | 103576-100 |
| Seahorse XF Calibrant | Agilent | 103059-000 |
| MitoStress Test kit | Agilent | 103015-100 |
| MitoTEMPO | SCBT | sc-221945 |
| Calcein AM | Sigma | C1359 |
| SYTOX AADvanced | Thermo Fisher | A35135 |
| CFSE Cell Division Tracker Kit | Biolegend | 423801 |
| Ficoll | | |

| Reagent/resource | Reference or source | Identifier or catalog number |
|---|---|---|
| SuperScript IV Reverse Transcriptase | Thermo Fisher | 18090050 |
| Random Hexamers | Qiagen | 79236 |
| Oligo-dT Primers | Qiagen | 79723 |
| miRneasy RNA extraction kit | Qiagen | 217084 |
| Caspase-Glo 3/7 Assay | Promega | G8090 |
| IFN-g ELISA | Biolegend | 430104 |
| GZMB ELISA | Biolegend | 439204 |
| PRF1 ELISA | Mabtech | 3465-1HP-2 |
| Glucose Uptake Glo Assay | Promega | J1342 |
| FCCP | Cayman Chemical | 15218IF |
| Poly-L-Lysine | Sigma | P4707 |
| Complete Protease Inhibitor Cocktail | Roche | 4693116001 |
| Protease Inhibitor Cocktail | Cell Signaling Technology | 5871 |
| Phosphatase Inhibitor Cocktail | Cell Signaling Technology | 5870 |
| Cell Lysis Buffer | Cell Signaling Technology | 9803 |
| Mini Protean TGX Western Blot Gels | Bio-Rad | 456-1094 |
| Clarity Western ECL | Bio-Rad | 1705061 |
| Horseradish peroxidase | Biolegend | 405210 |
| K562 cell line (human) | ATCC | CCL-243 |
| Panc-1 cell line (human) | DSMZ | ACC783 |
| **Software** | | |
| ImageJ | Open source | n/a |
| Leica Application Suite X | Leica | n/a |
| Graphpad Prism, V10.6.1 | Siemens AG | n/a |
| FlowJo | BD | n/a |
| Proteo Wizard | Proteo Wizard | n/a |
| MetaboAnalyst | MetaboAnalyst | n/a |
| Perseus | MaxQuant | n/a |
| Spectronaut | Biognosys | n/a |
| **Other** | | |
| Seahorse XF HS Mini Analyzer | Agilent | S7852A |
| Confocal Microscope SP8 | Leica | n/a |
| ECIS zeta | Applied biophysics | n/a |
| Vi-Cell XR Cell Counter | BD | 383556 |
| NanoDrop2000 | Thermo Fisher | ND-2000 |
| LightCycler 480 | Roche | 05015243001 |
| Facs Canto II | BD | 641889 |
| MacsQuant 10 | Miltenyi | 130-096-343 |
| AutoMACS Pro Separator | Miltenyi | 130-092-245 |
| UHPLC System | Nexera | n/a |

| Reagent/resource | Reference or source | Identifier or catalog number |
|---|---|---|
| TripleTOF 6600 | AB Sciex | n/a |
| FilterMax F3 | Molecular Devices | F3-001 |
| 8 Well microscope slides | Ibidi | 80841 |
| Trans-Blot Turbo Transfer System | Bio-Rad | 1704150 |

## Methods and protocols

### Study design and patient recruitment

Patients undergoing elective major surgery were enrolled. Surgical procedures consisted of cardiac surgery with the use of cardio-pulmonary bypass, off-pump cardiac surgery, major orthopedic surgery and minor orthopedic surgery. The study was approved by the local ethics committee of the Ludwig-Maximilians-University Munich (Nr. 17-241 and 22-0384) and sample size estimation was conducted in the respective ethics applications. Written informed consent was obtained from all patients. In addition to informed consent and the WMA Declaration of Helsinki, the experiments also confirmed to the principles set out in the Department of Health and Human Services Belmont Report.

Age <18 years, emergency surgery, pregnancy, known malignancies, acute inflammatory processes prior to surgery, previous organ transplantation, immune system dysfunction or immunosuppression, and preoperative steroid treatment were considered as exclusion criteria.

Anesthesia was performed following internal protocols of the clinic: no premedication was applied; general anesthesia was induced using sufentanil 0.5–1 μg/kg, propofol 2 mg/kg and rocuronium 0.5–1 mg/kg. Administration of 0.7–1.2 μg/kg/h sufentanil and either 1.5–2 vol% sevoflurane or 6–8 mg/kg/h propofol was used to maintain anesthesia. For intraoperative muscle relaxation, fractionated i.v. application of rocuronium was applied. Pressure-controlled mechanical ventilation was used for all patients after endotracheal intubation with tidal volumes of ~6–8 ml/kg body weight. Blood was obtained after induction of anesthesia (T1), at the end of the surgery (T2) and on the first postoperative day on the intensive care unit (T3). Heparinized blood was processed immediately after collection.

### Graphics

Figure 1A and synopsis were created with BioRender.com.

### Peripheral blood mononuclear cell isolation and T-cell stimulation

Peripheral blood mononuclear cells (PBMC) were isolated by density gradient centrifugation (Histopaque 1077, Sigma-Aldrich). A ViCell analyzer (Beckman Coulter) was used to assess cell number and viability. PBMC were suspended in RPMI 1640 (Invitrogen) supplemented with 200 mg/dl glucose, 10% heat-inactivated fetal calf serum (Biochrom), 1% HEPES (Sigma-Aldrich) and 1% L-glutamine (Life Technologies). PBMC were incubated at 37 °C and 5% $CO_2$ for 18 h. T-cell stimulation was performed using CD3/CD28 Dynabeads (Thermo Fisher Scientific) according to the manufacturer's instructions with a bead-to-cell ratio of 1:8.

### Impact of mitochondrial depolarization on T-cell immunity

To investigate the impact of impaired mitochondrial integrity on T-cell immunometabolism, PBMC were isolated, seeded and stimulated as described above. FCCP (Carbonyl cyanide-p-trifluoromethoxyphenylhydrazone) was added to a final well concentration of 20 nM. For vehicle controls (NC), equal amounts of JC1 assay buffer were used (FCCP and JC1 Assay Buffer: Cat. No. 701560, Cayman Chemical).

### Rescue of mitochondrial antioxidative capacity

To assess the immunometabolic impact of elevated antioxidant capacity, T2 PBMC were stimulated with anti-CD3/CD28 beads (Thermo Fisher) as described above and incubated with 1 μM MitoTEMPO (Cayman Chemical). Through the combination of the piperidine nitroxide TEMPO and the lipophilic cation triphenyl-phosphonium, MitoTEMPO can easily pass through membranes and accumulate in the mitochondria at high concentrations.

### CD8+ T-cell isolation

Prior to T-cell isolation, PBMC were harvested and CD3/CD28 Dynabeads were magnetically removed. Magnetic cell separation was used to isolate cytotoxic CD8+ T-cells (Human CD8 MicroBeads, #130-045-201 | CD8+ T Cell Isolation Kit, human, #130-096-495; Miltenyi Biotec) on an AutoMACS Pro Separator (#130-092-545, Miltenyi Biotec) according to the manufacturer's protocol.

### Next-generation Sequencing (NGS)

NGS was performed by Novogene Company Limited (Cambridge, UK) according to the company's specific protocol. In brief, sequencing libraries were generated using the QuantSeq 3' mRNA-Seq Library Prep Kit FWD for Illumina (Lexogen). The PCR Add-on Kit for Illumina (Lexogen) served to determine the optimal number of amplification cycles. Amplification was conducted according to the manufacturer's protocol. Quality and quantity of sequencing libraries were determined using the Quanti-iT PicoGreen dsDNA Assay Kit (Invitrogen) and the Bioanalyzer High Sensitivity DNA Analysis Kit (Agilent Technologies). Sequencing of libraries was performed on an Illumina HiSeq4000 sequencing machine (Illumina). Individually barcoded libraries were pooled and distributed across lanes of the same flow-cell aiming for approximately ten million paired-end reads per sample.

### Analysis of NGS data

Downstream analysis was performed using a combination of programs including STAR, HTseq, Cufflink and wrapped scripts. Alignments were parsed using the Tophat program and differential expressions were determined through DESeq2/edgeR. Gene Ontology (GO) and Kyoto Encyclopedia Of Genes And Genomes (KEGG) enrichment were implemented by the ClusterProfiler. Reference genome and gene model annotation files were downloaded from the genome website browser (NCBI/UCSC/Ensembl) directly. Indexes of the reference genome were built using STAR and paired-end clean reads were aligned to the reference genome using STAR (v2.5). HTSeq v0.6.1 was used to count the read numbers mapped of each gene. Next, fragments per kilobase per million mapped reads (FPKM) of each gene was calculated based on gene length and mapped read counts.

Differential expression analysis between two conditions/groups (two biological replicates per condition) was performed using the DESeq2 R package (2_1.6.3) and the edgeR R package (3.16.5) The resulting P values were adjusted using the Benjamini and Hochberg's approach for controlling the false-discovery rate (FDR). Genes with an adjusted P value <0.05 found by DESeq2 were assigned as differentially expressed. Prior to differential gene expression analysis, for each sequenced library, the read counts were adjusted by the edgeR program package through one scaling normalized factor.

To identify the correlation between differences in gene expression, samples were clustered using expression level FPKM to see the correlation using hierarchical clustering distance method with the function of heat map, SOM (Self-organization mapping) and means using silhouette coefficient to adapt the optimal classification with default parameter in R.

GO enrichment analysis of differentially expressed genes was implemented by the clusterProfiler R package, in which gene length bias was corrected. GO terms with corrected P value less than 0.05 were considered significantly enriched by differential expressed genes. Statistical enrichment of differential expression genes in KEGG pathways was assessed by the clusterProfiler R package.

### Cytotoxicity assay

Isolated primary human CD8+ T cells obtained at time points T1 or T2 were co-cultivated with calcein-prelabeled Panc-1 human pancreas ductal adenocarcinoma cells or calcein-labeled K562 lymphoblasts, using 8 μM calcein AM (#C1359, Sigma-Aldrich). Lysis-dependent increase of calcein fluorescence in the supernatant was quantified on a Filtermax F3 (excitation filter: 480 nm; emission filter: 520 nm|molecular devices). 1% Triton served as a positive control. Calculation of relative cell lysis capacity was performed using the formula [(test release − spontaneous release)/(maximum release − spontaneous release)] × 100.

For impedance-based assessment of cytotoxicity, $2 \times 10^5$ Panc-1 cells were seeded on gelatin precoated electric cell-substrate impedance sensing array, containing eight-wells with 40 gold electrodes per well (ECIS 8W10E + PET; Ibidi). Impedance was measured with multiple frequencies mode using the ECIS®Z system (Applied BioPhysics). Attachment of the cells to the well surface was indicated by impedance plateau generation. Subsequently, T1/T2 CTL were added in a 5:1 ratio and impedance drop was measured for 24 h. 1% Triton X-100 served as a positive control. The area under the curve (AUC) of the measured impedance at 4000 Hz was determined using Graphpad Prism 10. Relative cell lysis was evaluated using the formula (positive control AUC/test AUC) × 100. Target cell lines were authenticated by DSMZ.

For flow cytometric assessment of cytotoxicity, CTL obtained at time points T1 or T2 were co-cultivated with K562 lymphoblasts for 3 h in a ratio of 5:1. Afterwards cells have been stained using Sytoxx AADvanced and CD8 BV421 and cytotoxicity has been quantified on a MACSQuant 10 Flow Cytometer (Miltenyi Biotec).

### Caspase-3/7 activity assay

Caspase-3 and −7 activity was measured using the Caspase-Glo® 3/7 Assay (G8091, Promega Corporation), according to the manufacturer's instructions. Briefly, after co-incubation with CD8+ T cells, 5,000 K562 target cells were seeded in 100 μL of culture medium in white-walled 96-well plates. Subsequently,

100 µL of Caspase-Glo® 3/7 Reagent was added to each well. Plates were shaken at 400 rpm for 30 s and incubated at room temperature for 1 h. Luminescence was measured using a FilterMax F3 plate reader (Molecular Devices). Blanks containing culture medium and reagent without cells were included to determine background luminescence. Measurements were performed in triplicate.

### ELISA

To quantify cytokine secretion, enzyme-linked immunosorbent assay (ELISA) was performed according to the protocol of the respective manufacturer (IFNγ: #430104; Granzyme B: #439207; Biolegend, San Diego, CA, USA | Perforin: #3465-1HP-2, Mabtec). Absorbance was measured on a Filtermax F3 using a plate-specific standard curve.

### Oxygen consumption rate and extracellular acidification rate

Functional evaluation of mitochondrial oxidative phosphorylation and cellular glycolysis was conducted by extracellular flux analysis using a Seahorse HS mini (Agilent). About 150,000 T cells were plated on an eight-well HS mini plate (#103723-100, Agilent) precoated with poly-L-lysine (Biochrom, #L7240) in a final volume of 180 µl Assay Medium (containing Seahorse RPMI supplemented with 1 mM sodium pyruvate, 2 mM glutamine and 5.5 mM glucose). All experiments were conducted in duplicates or triplicates. Quantification of extracellular acidification rate (ECAR) and oxygen consumption rate (OCR) was performed in response to MitoStress Test (#103015-100, Agilent). To this end, final well concentrations of 1 µM Oligomycin, 1 µM FCCP/2.5 µM BAM15 and 0.5 µM rotenone/antimycin A were consecutively added through designated seahorse cartridge compound delivery ports.

### Quantification of cellular glucose uptake

Glucose uptake was measured using the Glucose Uptake Glo Assay (Promega) according to the manufacturer's instructions. Briefly, 150,000 cells were incubated with 1 mM 2-deoxyglucose (2-DG) for 10 min at room temperature. After incubation, 50 µl of each sample was transferred to a white luciferase plate. Control samples were prepared without 2-DG. Stop buffer, neutralization buffer, and luciferase detection reagent were added sequentially, and samples were incubated for 30 min at room temperature in the dark. Luminescence was measured using a SoftMax instrument, and raw data were exported for analysis. All samples were measured in duplicate.

### Mitochondrial membrane potential

Mitochondrial membrane potential ΔψM was assessed using the membrane permeable lipophilic cationic carbocyanine dye JC1 according to the manufacturer's protocol (#701560, Cayman Chemical). Data were acquired on a FACS Canto II flow cytometer (BD Biosciences, Franklin Lakes, USA). In mitochondria with intact membrane potential, ΔψM-dependent JC1 accumulation leads to the formation of J-aggregates, emitting red fluorescence (~590 nm). Depolarization of the mitochondrial membrane potential results in lower cellular concentrations of the dye, forming green fluorescent monomeric forms of JC1 (~529 nm). ΔψM is depicted as the mean fluorescence intensity ratio of red/green. The uncoupling agent FCCP (Carbonyl cyanide-p-trifluoromethoxy-phenylhydrazone), causing almost complete loss of ΔψM at doses of 20 µM, served as a negative control.

### Quantification of reactive oxygen species (ROS)

CellROX Green and MitoSOX red (C10492|M36008; Thermo Fisher Scientific) were used to quantify intracellular and mitochondrial ROS levels, respectively. Experiments were performed according to the manufacturer's instructions. Briefly, $2 \times 10^5$ cells were incubated in RPMI supplemented with 500 nM CellROX green for 30 min at 37 °C and 5% $CO_2$. For MitoSOX, cells were incubated in prewarmed HBSS supplemented with 0.2 µM Mito-SOX red for 15 min at 37 °C and 5% $CO_2$. Cells were immediately analyzed on a FACS Canto II flow cytometer (BD Biosciences). To estimate the peroxidase-mediated production of $H_2O_2$ in sorted cell populations, $10–20 \times 10^5$ cells have been seeded in a 96-well plate and incubated with medium:luminol in a 4:1 ratio at room temperature for 5 min. Wells with medium only and wells with luminol, horse reddish peroxidase and $H_2O_2$ served as a negative or positive control, respectively. Luminescence was measured using a SoftMax instrument.

### Antioxidant capacity

To evaluate cellular antioxidant capacity via quantification of intracellular reduced glutathione (GSH), $2 \times 10^5$ cells were incubated in PBS supplemented with 10 µM ThiolTracker (T10095, Thermo Fisher) according to the manufacturer's protocol. Data were acquired on a FACS Canto II flow cytometer.

### Analysis of mitochondrial mass

ΔψM-independent MitoTracker Green FM (#9074, Cell Signaling Technology) was used for flow cytometric determination of mitochondrial mass ($2 \times 10^5$ cells, 200 nM MitoTracker, incubation: 15 min, 37 °C in the dark). Mitochondrial mass was obtained by quantification of mean fluorescence intensity (MFI) green on a FACS Canto II flow cytometer (BD Biosciences).

### Flow cytometry

For extracellular CD8$^+$ T cell staining, $2 \times 10^5$ PBMC were preincubated with 2.5 µl Human TruStain FcX™ Fc Receptor Blocking Solution (#422302, BioLegend), followed by incubation with the designated antibody (PerCP anti-human CD8 (#344707 BioLegend), for a duration of 30 min on ice, protected from light. Intracellular assessment of Interferon-γ, Granzyme B and Perforin 1 was performed using FITC anti-human Interferon γ (#502506, BioLegend), BV421 anti-human Granzyme B (#396413, BioLegend) and APC anti-human Perforin 1 (#308111 BioLegend) after fixation and permeabilization using eBioscience™ Fixation/Permeabilization Concentrate, Diluent and Buffer (#00-5123-43 | #00-5223-56 | #00-8333-56, Invitrogen) as to the manufacturer's protocol.

Detection of MDSC was performed using $2 \times 10^5$ PBMC, preincubated with 2.5 µl Human TruStain FcX™ Fc Receptor Blocking Solution (#422302, BioLegend). Cells were stained with PE anti-human CD15, Pacific Blue anti-human HLA-DR, PerCP anti-human CD14, and FITC anti-human CD11b (#301905, #307623, #325631, #301329; BioLegend).

Quantification of T-cell activation and exhaustion marker (CD25, Ce38, CD69, PD1, TIM3, and LAG3) was performed on magnetically separated CTL using the antibodies specified in the reagent and tools table. About $2 \times 10^5$ CTL were incubation with the designated antibodies for a duration of 30 min on ice, protected from light.

For quantification of cytolytic granules exocytosis, detection of plasma membrane incorporation of CD107a on CTL was performed. CTL were incubated as described above. After 20 h, CD107a PE (#328608, Invitrogen) was added and cells were stimulated using 1x Cell Activation Cocktail (#423301, Biolegend). After 1 h, 1:1000 Brefeldin A (#00-4506-51, Invitrogen) was added and cells were incubated for an additional 3 h at 37 °C. Afterwards, cells were washed and fixation, permeabilization and intracellular staining was performed as described above. For all extracellular stainings, antibodies have been diluted 1:100, for intracellular staining, a 1:40 dilution has been used.

Cells were analyzed on a FACS Canto II (BD Biosciences, USA) and on a MACSQuant 10 (Miltenyi, Germany). Data analyses were performed using the software FlowJo v10 (FlowJo, Ashland, USA).

### Isolation of mitochondria

Mitochondria were isolated from primary human $CD8^+$ T cells using the Mitochondria Isolation Kit, human (#130-094-532, Miltenyi Biotec) according to the manufacturer's protocol. In brief, isolated $CD8^+$ T cells cells were resuspended in lysis buffer and homogenized using a 26G needle. Homogenized mitochondria were magnetically labeled with Anti-TOM22 Microbeads. Magnetic separation was performed on an AutoMACS Pro Separator.

### Confocal microscopy

For microscopic imaging, CTL were incubated as described above. Subsequently, cells were harvested and seeded at a density of $0.25 \times 10^6$ in eight-well microscope slides (#80841, ibidi) coated with 0.01% poly-L-Lysine (#P4707, Sigma-Aldrich). Microscopy slides were centrifuged at $300 \times g$ for 5 min and incubated for 10 min at 37 °C to allow attachment of cells. After fixation with ice-cold 100% methanol (incubation for 20 min. at −20 °C), cells were washed with PBS and stained with MitoTracker Deep Red (#8778, Cell Signaling Technologies) according to manufacturer's instructions, fixed with ice-cold methanol at −20 °C for 20 min. Thereafter, cells were washed, incubated with 5% BSA for 1 h and immunostained with 1:50 anti-CD3 primary antibody (#SC20047, Santa Cruz Biotechnologies), and subsequent 1:2000 anti-mouse Alexa Fluor 488 antibody (#A21042, Invitrogen). Chamber inserts were removed and slides were mounted with mounting medium (ibidi, #50001). Confocal images were recorded on a Leica SP8 microscope based on an upright DM6 stand using solid state laser excitation at 488 and 638 nm with a Leica DFC365FX with Sony ICX285 interline CCD chip. Hybrid photo detectors with detection bands of 500–550 nm and 650–700 nm were used. Images were acquired with a 63×1.4 or a 40x PL APO oil immersion objective at a scan speed of 200 or 400 Hz with a pixel size of of 0.075 μm and a frame size of $1024 \times 1024$ (for measuring mitochondrial length) or with a pixel size of 0.4 μm and a frame size of $512 \times 512$ (for determining mitochondrial translocation to the immune synapse). For each image, 3D stacks were recorded (Z-Stack size: 0.3–0.5 μm). Analysis was performed on projections of the image stacks. For image visualization, Leica Application Suite X (LASX) was used. Data has been uploaded to the BioStudies depository (www.ebi.ac.uk/biostudies, accession no. S-BIAD2197).

### Evaluation of mitochondrial translocation to the immunological synapse and mitochondrial length

Mitochondrial translocation to the immunological synapse of cytotoxic T cells was determined by (1) approximation of CD3 (T-cell Receptor, TCR) and MitoTracker (mitochondria) and by (2) concentration of mitochondria to the immunological synapse in contrast to dispersed mitochondrial staining throughout the cell. CTL were directly isolated after blood withdrawal, activated using CD3/CD28 dynabeads at a bead:cell ratio of 1:8 and incubated at 37 °C and 5% $CO_2$ for 20 h. Immediately prior to fixation and staining as described above, dynabeads were magnetically removed. For each patient and each time point, at least three Z-stacks with maximum projection were evaluated by three individual scientists. Mean values of polarized/non-polarized cells per image were then used for statistical analysis. Evaluation of mitochondrial length has been performed as previously described (Triolo et al, 2023). In short, after acquisition of confocal images as specified above, Fiji software was used to generate 3D image projections, and the line function was applied to draw individual lines across each mitochondrion identified. Mitochondrial count per cell and mean mitochondrial area per cell were additionally quantified using the Mitochondrial Analyzer ImageJ plugin as previously described (Chaudhry et al, 2020).

### LC-MS/MS-based untargeted metabolomics

For analyses of serum metabolites, blood was withdrawn after induction of anesthesia (T1), at the end of surgery (T2) and on the first postoperative day (T3). S-Monovette® Serum tubes (Cat.# 02.226.160, Sarstedt, Germany) were stored at an upright position for 30 min at room temperature to allow coagulation. Serum was collected at 15 °C, 2750 g/min for 10 min., immediately frozen and stored at −80 °C until analysis.

Untargeted metabolome analysis was carried out on a Nexera UHPLC system connected to a Q-TOF mass spectrometer (TripleTOF 6600, AB Sciex). Frozen samples were thawed, 25 μl of each sample were mixed with 125 μl MeOH and incubated for 10 min at 10 °C with shaking at 1000 rpm. After 5 min sonication, samples were centrifuged for 10 min at 13000 rpm at 10 °C. Chromatographic separation was achieved by using a HILIC UPLC BEH Amide $2.1 \times 100$, 1.7 μm column with 0.4 mL/min flow rate. The mobile phase consisted of 5 mM ammonium acetate in water (eluent A) and 5 mM ammonium acetate in acetonitrile/water (95/5, v/v) (eluent B). The following gradient profile was used: 100% B from 0 to 1.5 min, 60% B at 8 min and 20% B at 10 min to 11.5 min and 100% B at 12 to 15 min. About 5 μL aliquots per sample were injected into the UHPLC-TOF-MS. The autosampler was cooled to 10 °C and the column oven heated to 40 °C. A quality control (QC) sample—pooled from all samples—was injected after every ten samples. MS settings in the positive mode were: Gas 1 55, Gas 2 65, Curtain gas 35, Temperature 500 °C, Ion Spray Voltage 5500, and declustering potential 80. Changes for the negative mode were ion spray voltage −4500 and declustering potential −80. For both modes, the mass range of the TOF-MS scans were 50–2000 m/z and the collision energy was ramped from 15 to 55 V.

The "msconvert" from Proteo Wizard (Kessner et al, 2008) served to convert raw files to mzXML (de-noised by centroid peaks). Data processing and feature identification was performed via the bioconductor/R package xcms (Smith et al, 2006). More specifically, the matched Filter algorithm was used to identify peaks (full width at half maximum set to 7.5 s). Peaks were then grouped into features using the "peak density" method (Smith et al, 2006). Area under the peak was integrated to represent the abundance of features. The retention time was adjusted based on the peak groups

presented in most samples. To enable feature annotation to metabolites, the exact mass and MS2 fragmentation pattern of the measured features were compared to the records in HMBD (Wishart et al, 2018) and the public MS/MS spectra in MSDIAL (Tsugawa et al, 2015), referred to as MS1 and MS2 annotation, respectively. All samples were pooled to generate one reference sample, which was measured seven times throughout the measurements. The reference samples were used to control the drifting effect of the mass spectrometer's signal, using a method adapted from EigenMS (Karpievitch et al, 2014). In detail, we used singular value decomposition of the reference measurements to estimate the variance contributed by the artifact reasons. These were removed from all measurements to get the normalized metabolite intensity matrix. Missing values were imputed with half of the limit of detection (LOD) methods. Paired $t$-test served to compare the features' intensity from T1 to T2. Significant enrichment of the metabolites in the groups was defined as a mean difference (mean diff.) >1.5 with a false-discovery-rate FDR <0.05. When MS2 spectra were acquired in our experiments, our MS2 fragmentation pattern was manually reviewed and compared to records in the public database or previously measured reference standards to evaluate the correctness of the annotation.

For functional analysis of untargeted metabolomics data, the online tool MetaboAnalyst was used as previously described (Pang et al, 2022). Peak intensities were uploaded including retention time and numeric mass (M/z) values. Features with a constant or single value across samples were deleted. Missing values were replaced by 1/5 of min positive values of their corresponding variables. A 25% interquartile range variance filter was applied for data filtering. The Mummichog 2.0 algorithm was used with $p < 0.05$ as a cut-off. The MFN human pathway library served for identification of significantly enriched pathways with at least two entries per pathway. Knowledge-driven integration of metabolomic and proteomic data was performed with the online tool MetaboAnalyst. Compounds and proteins with significant changes have been uploaded and joint pathway analysis has been calculated using all KEGG pathways. Tight integration was applied by combining queries.

### Serum proteomics

Proteins from 1 μl sample were digested using the autoSP3 protocol (Müller et al, 2020) on an Agilent Bravo liquid handling system in a 96-well format. Samples were randomized on the plate. Briefly, samples were diluted in 100 mM ammonium bicarbonate including complete protease inhibitor cocktail (Roche). Proteins were denatured, reduced and alkylated by addition of a 4x buffer including TCEP, CAA, and SDS and heated at 95 °C for 10 min. A 1:1 mixture of hydrophobic and hydrophilic magnetic beads was added (Sera-Mag, Cytiva) and acetonitrile (ACN) filled up to a final concentration of 50% (v/v). Proteins were allowed to bind to beads while shaking at room temperature. Beads were washed twice with 80% ethanol (v/v) and once with ACN. Bead pellets were resuspended in 100 mM ammonium bicarbonate and trypsin protease (Sigma-Aldrich) added at an enzyme:protein ratio of 1:50 (w/w). Digestion was performed overnight at 37 °C. Peptides were acidified to a final concentration of 1% TFA (v/v). 0.5% of this solution was loaded on Evotips (Evosep). As quality control, four pooled serum samples (Pat1_T2, Pat2_T1, Pat3_T3, Pat7_T2, Pat8_T1, Pat9_T3, Pat13_T2, Pat14_T1, Pat15_T3, Pat19_T1, and Pat20_T2) were distributed on different positions on the plate.

LC-MS analysis was performed on a Evosep One LC (Evosep) using the 30 spd method with a 15 cm column (150 μm ID, 1.5 μm, PepSep). Eluted peptides were electrosprayed into an Exploris 480 mass spectrometer (Thermo Fisher Scientific). The mass spectrometer was operated in data-independent acquisition mode with a full scan with resolution R = 120,000 at m/z 200, m/z range between 380 and 980, 300% AGC target, 100 ms max. injection time; followed by 30 DIA windows at 20 $m/z$ with an overlap of 1 $m/z$ at resolution R = 30000 at $m/z$ 200, 3000% AGC target, auto IT in centroid mode.

Data was analyzed by Spectronaut (version 18.0.230605.50606) in directDIA+ mode using the database uniprot_sprot_2022-01-07_HUMAN (20375 protein entries) downloaded in the software. Two missed cleavages were chosen, and Carbamidomethyl (C) as fixed modification as well as Acetyl (protein N-term) and Oxidation (M) as variable modifications. Samples were normalized and identified (Qvalue) set for precursor filtering.

### mRNA quantification

Quantification of mRNA expression was performed on a Light-Cycler 480 (Roche) as described in detail previously (Möhnle et al, 2018; Hirschberger et al, 2019) in accordance with the MIQUE guidelines. In brief, the miRNeasy RNA Isolation Kit (#217004, Qiagen, Hilden, Germany) with on-column DNA digestion served for RNA isolation, which was performed as per the manufacturer's instructions. Quantity and purity of RNA were assessed on a NanoDrop2000 spectrophotometer (Thermo Fisher Scientific). First-strand cDNA was synthesized with equal amounts of RNA using a Superscript III reverse transcriptase (Invitrogen), oligo (dT) primers and random hexamers (Qiagen). Ribosomal Protein L13a (RPL13A) and TATA box binding protein (TBP) served as reference genes in all experiments (Ledderose et al, 2011). Primers, probes and identifiers of multiplexing assays are given in Table EV1. Duplicate values were used for all analysis. The second derivative maximum method was applied to define quantification cycles by the LightCycler software. A Quantification cycle (Cq) cut-off was set for Cq 35; values beyond this cut-off were considered as unspecific.

### Immunoblot

Cells were lysed in cell lysis buffer (#9803, Cell Signaling Technology) supplemented with a protease and phosphatase inhibitor (#5871 | #5870, Cell Signaling Technology). Equivalent protein concentrations were loaded onto polyacrylamide stacking gels (Mini-PROTEAN TGX 4-20%, #456-1094, Bio-Rad). After transfer, protein was blotted with 1:400 Total OXPHOS Human Antibody Cocktail (#ab110411, Abcam), 1:500 pDRP1, 1:500 DRP1. 1:1000 OPA1, and 1:1000 MFN2 antibodies (#3455S, #5391, #11925S, #67589, Cell Signaling Technology). As a loading control, 1:2000 beta-actin antibody (#5125, Cell Signaling Technology) was used. For mitochondrial isolates, 1:800 HSP60 (#4870, Cell Signaling Technology) served as a loading control.

### Statistical analyses

Statistical analysis was performed using GraphPad Prism 10 (GraphPad Software, Inc., USA). Paired $t$-test or Wilcoxon matched-pairs signed-rank test, as appropriate, served for comparisons. Normal distribution was tested using the D'Agostino & Pearson and Shapiro–Wilk test. If not stated otherwise, data are

**The paper explained**

**Problem**

Postoperative complications such as infections and wound healing disorders are common after major surgery. Up to 35% of patients suffer from nosocomial infections that delay recovery, and increase both mortality and overall healthcare costs. This is primarily caused by suppression of the adaptive immune system, particularly affecting cytotoxic T cells (CTL). The mechanism underlying this immunosuppression is unclear, and no therapeutic options are available.

**Results**

After major surgery, CTL exhibit a strongly reduced cytotoxic capability and their cytokine secretion is almost disrupted due to a blockade of cytolytic granules exocytosis. We found that this dysfunction is driven by systemic oxidative stress in the blood of surgical patients. We identified transiently emerging MDSC as the predominant source of this oxidative stress through exacerbated production of reactive oxygen species (ROS). These ROS overwhelm CTL antioxidant defenses and substantially damage CTL mitochondria. This results in impaired mitochondrial function with cell-intrinsic ROS production, leading to compromised ATP production and diminished CTL effector functions. Importantly, treatment with a mitochondria-targeted antioxidant (MitoTempo) improved mitochondrial integrity and CTL function in this model.

**Impact**

We uncovered a new mechanism of postoperative immune suppression, where MDSC-derived oxidative stress induces mitochondrial damage in CTL, creating an immunometabolic paralysis of CTL. Targeting oxidative stress with specific antioxidants may help restore CTL immunometabolism in surgical patients. These findings open up new therapeutic strategies that could lower the risk of infections, improve recovery, and reduce healthcare costs of patients after major surgery.

depicted as mean ± SEM or as box plots with mean, median, twenty-fifth and seventy-fifth percentiles and range, with dots indicating individual values. $*p < 0.05$, $**p < 0.01$, and $***p < 0.001$. Biological replicates and statistical parameters are reported in the figure legends.

## Data availability

The datasets produced in this study are available in the following databases: Transcriptomics: Array Express, E-MTAB-14974, https://www.ebi.ac.uk/biostudies/ArrayExpress/studies/E-MTAB-14974?query=E-MTAB-14974. Proteomics: ProteomeXchange Consortium via the PRIDE partner repository, PXS060723, https://www.ebi.ac.uk/pride/archive/projects/PXD060723/. Metabolomics: MassIVE Repository, MSV000097035, https://massive.ucsd.edu/ProteoSAFe/private-dataset.jsp?task=b04359ba2b2c4278a344d855d5d1a85b. Microscopy: BioImage Archive, S-BIAD2197, https://www.ebi.ac.uk/biostudies/bioimages/studies/S-BIAD2197.

The source data of this paper are collected in the following database record: biostudies:S-SCDT-10_1038-S44321-025-00324-1.

## Peer review information

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

## Acknowledgements

We thank Katja Gieseke, Florian Gosselin, Bärbel Reincke, Gudrun Prangenberg, and Gabriele Gröger for their excellent technical assistance. We thank the Core Facility Bioimaging and the Protein Analysis Unit of the Biomedical Center of the LMU for providing excellent assistance in acquisition of microscopy images and proteomics, respectively. This work has been funded by the Deutsche Forschungsgemeinschaft (#447514737, S.K.; #542908663, MH), by the Friedrich-Baur-Stiftung (#22/22, M.H.) and the Else Kröner-Fresenius Stiftung (2021_EKEA.19, M.H.). S.H. ("FöFoLe +", #024, LMU Munich, DFG) and M.H. (Program in Vascular Medicine, PRIME, funded by the Deutsche Forschungsgemeinschaft #413635475) have been supported by the Medical and Clinician Scientist Program of the LMU Munich. The TripleTOF 6600 mass spectrometer was funded in part by the German Research Foundation (INST 95/1434-1 FUG).

## Author contributions

**Simon Hirschberger**: Conceptualization; Resources; Data curation; Formal analysis; Supervision; Validation; Investigation; Visualization; Methodology; Writing—original draft; Project administration; Writing—review and editing. **Martin B Müller**: Resources; Formal analysis; Investigation; Methodology. **Hannah Mascolo**: Investigation. **Melissa Seitz**: Investigation. **Stefan Nibler**: Investigation. **David Effinger**: Investigation; Visualization; Methodology. **Kun Lu**: Resources. **Joscha Büch**: Resources. **Martin Bender**: Resources. **Tobias Kammerer**: Investigation. **Sven Peterß**: Resources. **Karin Kleigrewe**: Resources; Investigation; Methodology. **Miriam Abele**: Resources; Investigation; Methodology. **Teresa Barth**: Resources; Investigation; Methodology. **Olga Kushnir**: Investigation. **Axel Imhof**: Resources; Investigation; Methodology. **Steffen Dietzel**: Resources; Investigation; Methodology. **Bernd Wegener**: Resources. **Ralf Sowa**: Investigation. **Frank Vogel**: Resources. **Peter Lamm**: Resources. **Roland Tomasi**: Resources. **Kristian Unger**: Investigation. **Markus Sperandio**: Resources. **Erich Kilger**: Resources. **Simone Kreth**: Conceptualization; Resources; Data curation; Supervision; Funding acquisition; Writing—original draft; Project administration; Writing—review and editing. **Max Hübner**: Conceptualization; Resources; Data curation; Formal analysis; Supervision; Funding acquisition; Validation; Investigation; Visualization; Methodology; Writing—original draft; Project administration; Writing—review and editing.

Source data underlying figure panels in this paper may have individual authorship assigned. Where available, figure panel/source data authorship is listed in the following database record: biostudies:S-SCDT-10_1038-S44321-025-00324-1.

## Funding

## Disclosure and competing interests statement

The authors declare no competing interests.

# Expanded View Figures

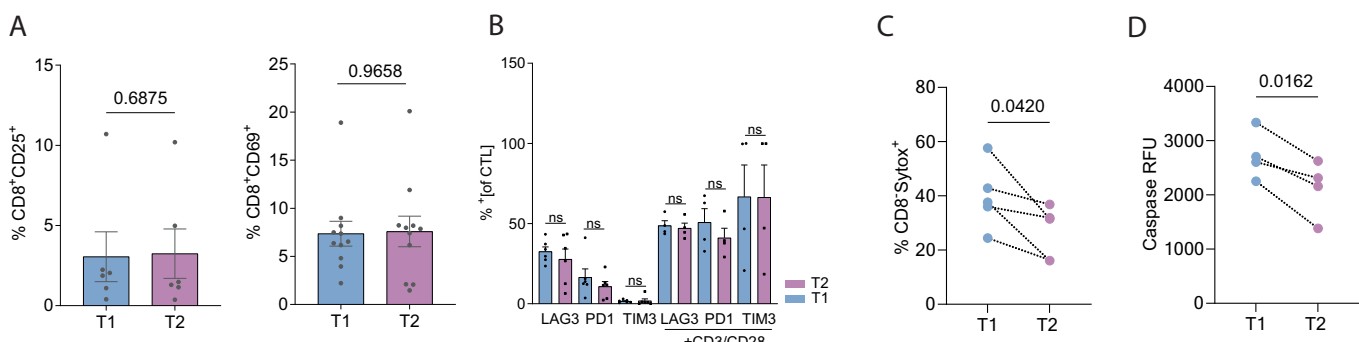

**Figure EV1. Post surgical immunosuppression (PITC) of CTL.**

(**A**) Flow cytometric analysis (FCA) of CD25 (left) and CD69 (right) staining and quantification in CTL, $n = 6/11$ individual patients. (**B**) FCA of lymphocyte-activation gene 3 (LAG3), programmed cell death protein 1 (PD1), and T-cell immunoglobulin and mucin-domain containing-3 (TIM3) staining and quantification as indicated, $n = 6/4$ (not activated/+CD3/CD28) individual patients. *P* values (left to right): 0.2842, 0.2188, 0.6875, 0.5494, 0.1250, and >0.9999. (**C**) Quantification of relative cytotoxicity using a flow cytometric killing assay, $n = 5$ individual patients. (**D**) Quantification of caspase-3 activity reporter using a CTL-K562 killing assay, $n = 4$ individual patients. Data were represented as mean ± SEM with dots indicating individual values (**A**, **B**) or as individual data points (**C**, **D**). *P* values: as indicated, paired *t*-test or Wilcoxon matched-pairs signed-rank test, as appropriate.

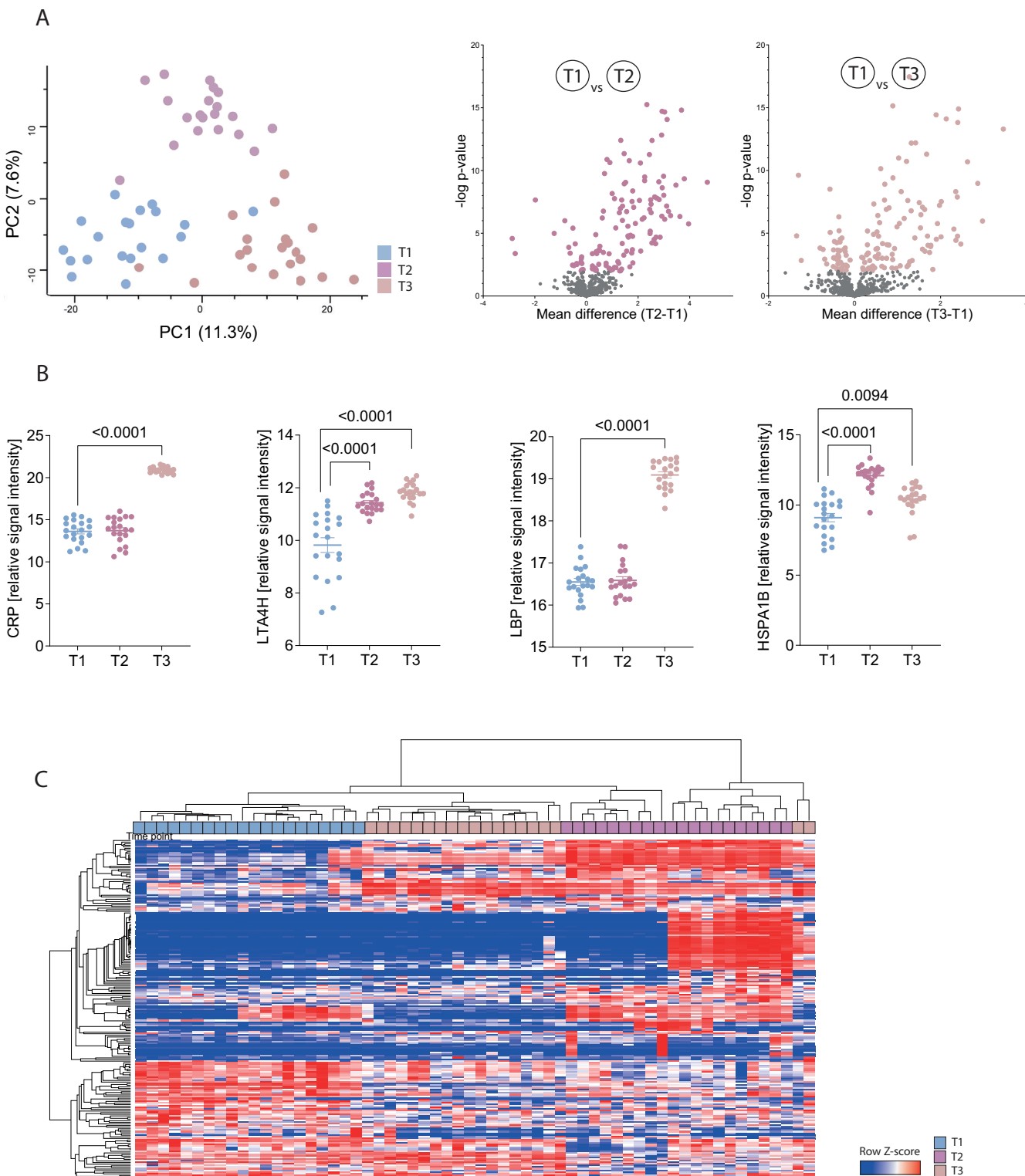

**Figure EV2.  Serum proteomics and metabolomics reveal an oxidative stress signature after major surgery.**

(A) Principal component analysis depicting proteomic profiling results during the course of major surgery, time points as indicated (left) and Volcano plots for protein abundance of comparisons as noted on the top. Proteins with $-\log p$ value >2 are colored, $n = 20$ individual patients. (B) Relative signal intensity of serum proteins during the course of major surgery as indicated, $n = 20$ individual patients. Mean ± SEM with dots representing individual values, $p$ values as indicated, paired $t$-test. (C) Heat map depicting quantity of significantly differentially expressed serum metabolites. Raw $z$-scores and time points as indicated, $n = 20$ individual patients.

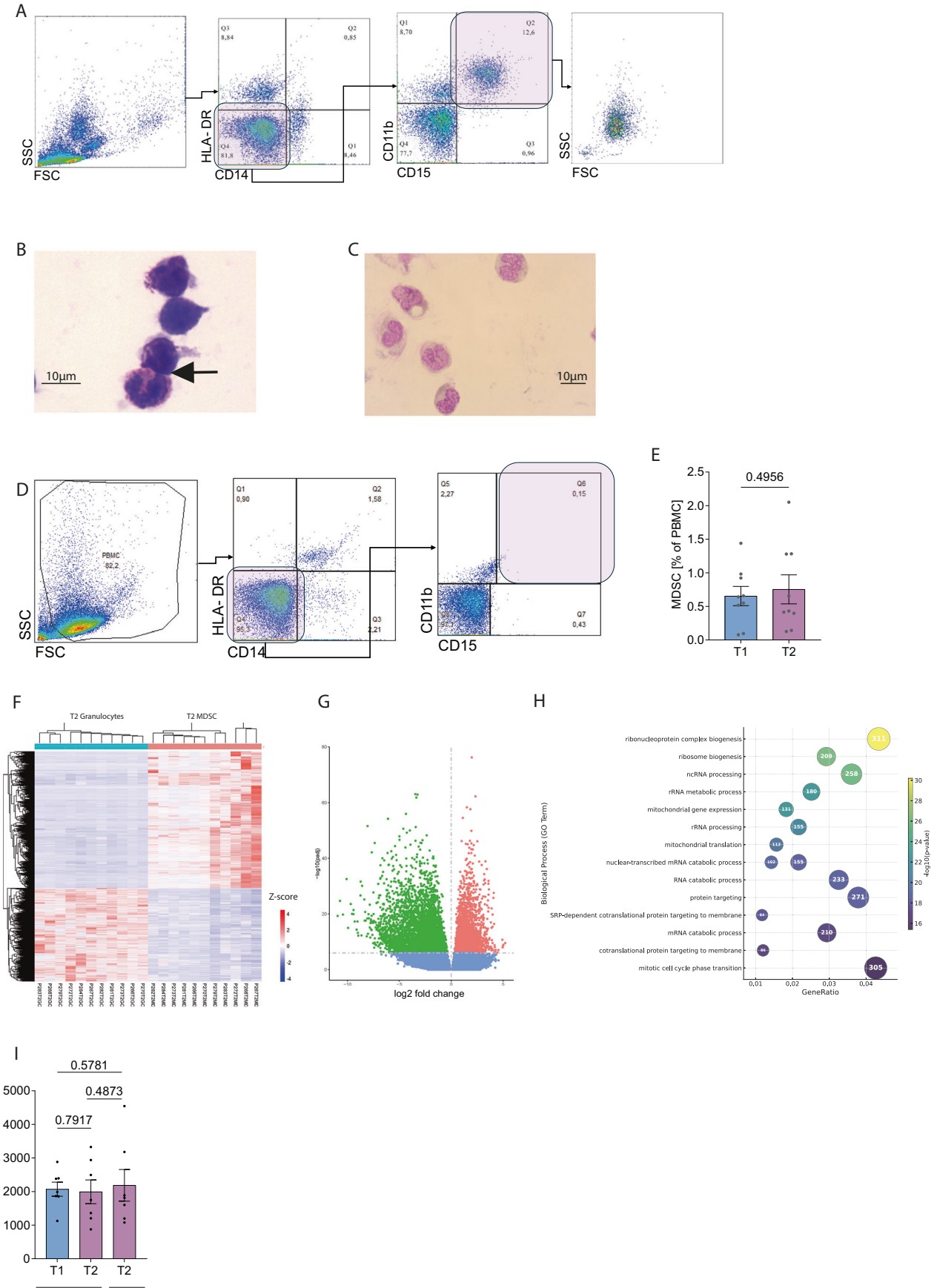

◀ **Figure EV3.   Transiently emerging myeloid-derived suppressor cells.**

(A) Flow cytometric gating strategy for immune phenotyping of CD14⁻HLA⁻DR-CD11b⁺CD15⁺ MDSC derived from patients during the course of major surgery. Representative scatter plots from a postoperative (T2) sample, as depicted in Fig. 3C. (B) Representative May–Grünwald–Giemsa staining of PBMC after major surgery. Transiently present cells depicting a granulocytic morphology (arrows). (C) Representative May–Grünwald–Giemsa staining of isolated MDSC, extracted from PBMC after major surgery using CD15 microbeads. (D) Flow cytometric analysis and immune phenotyping of PBMC from patients with minor surgery with gating strategy for detection of CD14⁻HLA⁻DR-CD11b⁺CD15⁺ MDSC. (E) Quantification of CD14⁻HLA⁻DR-CD11b⁺CD15⁺ MDSC as % of PBMC, derived from patients pre (T1) and post (T2) minor surgery, $n = 10$. (F) Heat map depicting quantity of all significantly differentially expressed transcripts of T2 MDSC vs T2 Granulocytes ($p < 0.0001$). Each pair of MDSC/ granulocytes derived from the same patient. Red color indicates an upregulation, and downregulated genes are indicated by blue color. (G) Volcano plots visualizing differential gene expression T2 MDSC/ T2 granulocytes. Log2 fold-changes (x-axis) and −log10 adjusted $p$ values (y-axis) are shown for each gene. The dashed horizontal line depicts the adj. $p$ value threshold of $p < 0.0001$, significantly regulated genes in red (upregulated in MDSC) and green (downregulated in MDSC). (H) Gene Ontology (GO) analysis of differentially expressed pathways in T2 MDSC/ T2 Granulocytes. The gene counts are expressed by the numbers within the bubbles. (I) Flow cytometric analysis and quantification of intracellular ROS as indicated by CellROX green staining in sorted cell populations, time points as stated, $n = 6$ individual patients. Data were represented as mean ± SEM with dots representing individual values. $P$ values as indicated, paired $t$-test.

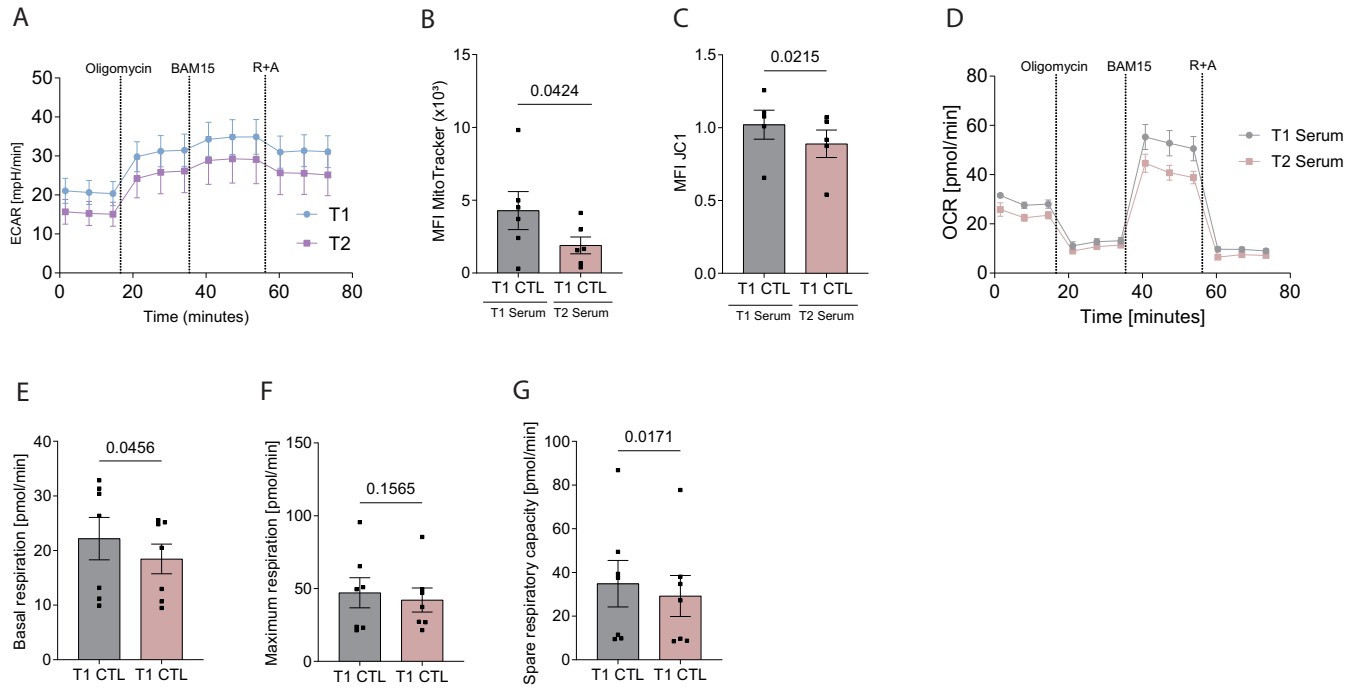

**Figure EV4. T1 CTL cultivated with T2 MDSC or T2 serum mimic phenotype of T2 CTL mitochondrial damage.**

(A) Pooled seahorse ECAR plot, mean ± SEM from five individual patients. (B–G) CTL of patients before major surgery (T1) co-cultivated with 20% serum from patients before (T1) or after (T2) major surgery. (B) FCA of MitoTracker green staining and quantification, $n = 6$ individual patients. (C) FCA of JC1 staining and quantification of red/green fluorescence ratio, $n = 5$ individual patients. (D) Representative seahorse OCR plot. (E–G) Seahorse quantification of (E) basal OCR, (F) maximal OCR, and (G) spare respiratory capacity in CTL, $n = 7$ from three individual patients. Data were represented as mean ± SEM with dots representing individual values. *P* values as indicated, paired *t*-test or Wilcoxon matched-pairs signed-rank test, as appropriate.

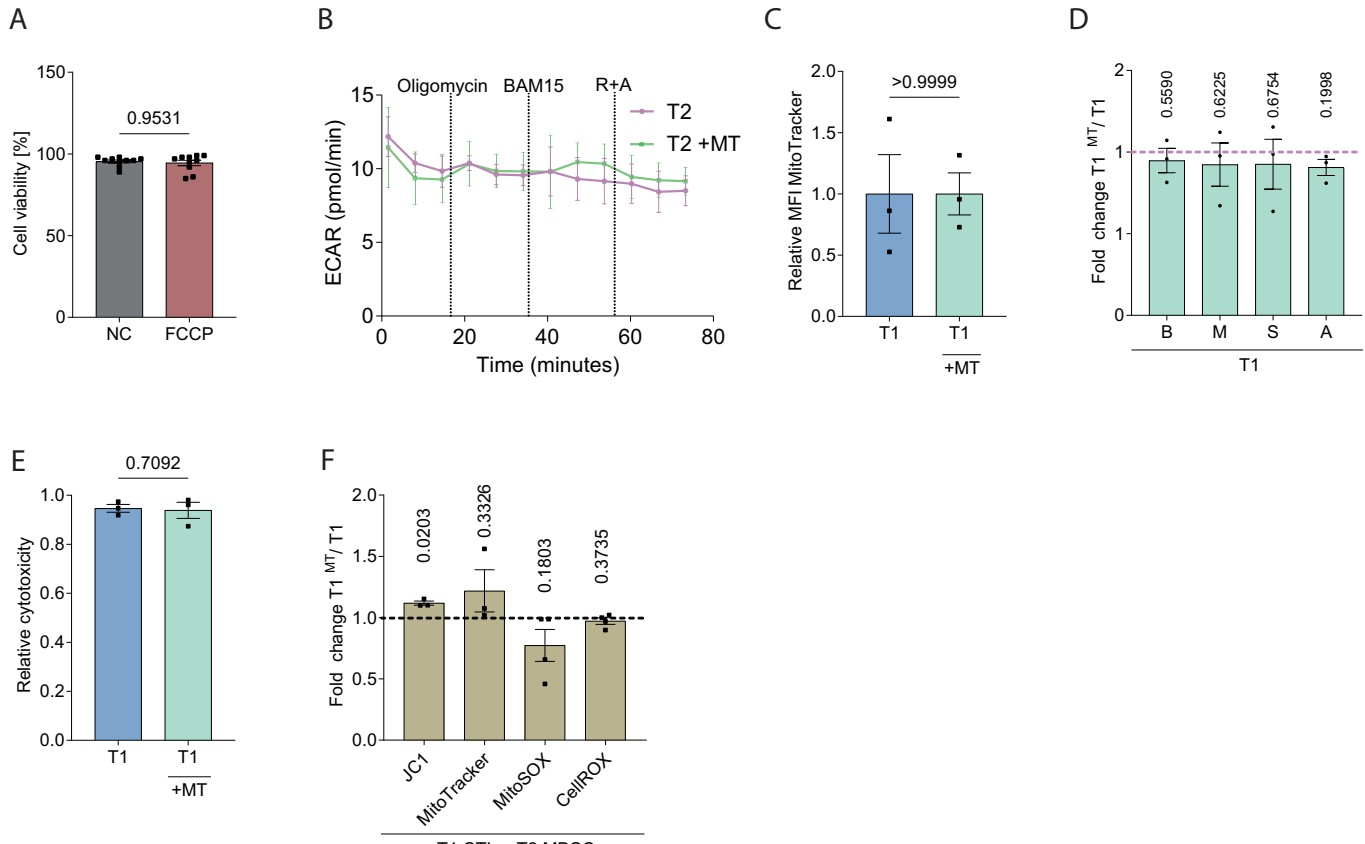

**Figure EV5. No impact of FCCP on cell viability and no impact of mitochondrial antioxidative treatment using MitoTEMPO (MT) on T1 CTL immunometabolism.**

(A) Cell viability of CTL treated with low-dose FCCP, $n = 10$, p value as indicated. (B) Pooled seahorse ECAR plot, mean ± SEM from six individual patients. (C) FCA of MitoTracker green and quantification in CTL, $n = 3$ individual patients. (D) Fold change of basal OCR (B), maximum OCR (M), spare respiratory capacity (S), and ATP production (A), $n = 3$ individual patients. (E) Quantification of relative CTL cytotoxicity using a flow cytometric killing assay, $n = 3$ individual patients. (F) Fold change of mitochondrial parameters as indicated comparing MT treated T1 CTL co-cultivated with MDSC versus untreated T1 CTL co-cultivated with MDSC, $n = 4/3/3/4$. Data were represented as mean ± SEM. P values as indicated, one-sample $t$-test (D) paired $t$-test or Wilcoxon matched-pairs signed-rank test, as appropriate.

