## [Peer Review File · EMBO Molecular Medicine]

Mitochondrial damage drives T-cell immunometabolic paralysis after major surgery

Simon Hirschberger, Martin Müller, Hannah Mascolo, Melissa Seitz, Stefan Nibler, David Effinger, Kun Lu, Joscha Büch, Martin Bender, Tobias Kammerer, Sven Peterß, Karin Kleigrew, Miriam Abele, Teresa Barth, Olga Kushnir, Axel Imhof, Steffen Dietzel, Bernd Wegener, Ralf Sowa, Frank Vogel, Peter Lamm, Roland Tomasi, Kristian Unger, Markus Sperandio, Erich Kilger, Simone Kreth, and Max Hübner

Corresponding author: Simone Kreth (simone.kreth@med.uni-muenchen.de)

Review Timeline:

Submission Date:	13th May 25
Editorial Decision:	13th Jun 25
Revision Received:	19th Aug 25
Editorial Decision:	17th Sep 25
Revision Received:	29th Sep 25
Accepted:	7th Oct 25

Editor: Zeljko Durdevic

Transaction Report:

13th Jun 2025

Dear Prof. Kreth,

Thank you for the submission of your manuscript to EMBO Molecular Medicine. We have now received feedback from the three reviewers who agreed to evaluate your manuscript. As you will see from the reports, all three referees recognize potential interest of the study, but they also raise serious and partially overlapping concerns that should be addressed in a major revision. If you would like to discuss further the points raised by the referees, I am available to do so via email or video. Let me know if you are interested in this option.

We would welcome the submission of a revised version within three months for further consideration. Please let us know if you require longer to complete the revision.

I look forward to receiving your revised manuscript.

Yours sincerely,

Zeljko Durdevic

Zeljko Durdevic
Senior Editor
EMBO Molecular Medicine

We require:

- 1) A .docx formatted version of the manuscript text (including legends for main figures, EV figures and tables). Please make sure that the changes are highlighted to be clearly visible.
- 2) Individual production quality figure files as .eps, .tif, .jpg (one file per figure). For guidance, download the 'Figure Guide PDF': (<https://www.embopress.org/page/journal/17574684/authorguide#figureformat>).
- 3) A .docx formatted letter INCLUDING the reviewers' reports and your detailed point-by-point responses to their comments. As part of the EMBO Press transparent editorial process, the point-by-point response is part of the Review Process File (RPF), which will be published alongside your paper.
- 4) A complete author checklist, which you can download from our author guidelines (<https://www.embopress.org/page/journal/17574684/authorguide#submissionofrevisions>). Please insert information in the checklist that is also reflected in the manuscript. The completed author checklist will also be part of the RPF.
- 5) Please note that all corresponding authors are required to supply an ORCID ID for their name upon submission of a revised manuscript.
- 6) It is mandatory to include a 'Data Availability' section after the Materials and Methods. Before submitting your revision, primary

datasets produced in this study need to be deposited in an appropriate public database, and the accession numbers and database listed under 'Data Availability'. Please remember to provide a reviewer password if the datasets are not yet public (see <https://www.embopress.org/page/journal/17574684/authorguide#dataavailability>).

12) Author contributions: You will be asked to provide CRediT (Contributor Role Taxonomy) terms in the submission system. These replace a narrative author contribution section in the manuscript.

13) A Conflict of Interest statement should be provided in the main text.

14) Every published paper now includes a 'Synopsis' to further enhance discoverability. Synopses are displayed on the journal webpage and are freely accessible to all readers. They include a short stand first (maximum of 300 characters, including space) as well as 2-5 one-sentences bullet points that summarizes the paper. Please write the bullet points to summarize the key NEW findings. They should be designed to be complementary to the abstract - i.e. not repeat the same text. We encourage inclusion of key acronyms and quantitative information (maximum of 30 words / bullet point). Please use the passive voice. Please attach

these in a separate file or send them by email, we will incorporate them accordingly.

15) Include a Reagents and Tools Table as part of the Methods section, which can be downloaded from our author guidelines (<https://www.embopress.org/page/journal/17574684/authorguide#structuredmethods>)

***** Reviewer's comments *****

Referee #1 (Comments on Novelty/Model System for Author):

Hirschberger et al propose that that following surgery T cells undergo metabolic dysfunction they propose that ROS derived from myeloid suppressor cells impaired mitochondrial function in T cells and influence T cell activity. They call this immune paralysis. Overall this work does offer new insights into mechanisms driving immune dysfunction following surgery. They suggest that ROS levels are contributing to T cell dysfunction and that T cell function may be partially restored by antioxidant activity. Overall, the manuscript is well written and the data are rigorous there are several revisions that would improve the clarity and quality of the manuscript. Further there are alternate interpretations of their data that should be considered.

Critique:

General: There is lack of consistency in figures with flow histograms accompanied by bar/scatter graphs. Sometimes left to right in the bar graph matches top to bottom (Fig 4 a-d) and sometimes it is the opposite (Fig 1d,h, 4e). Formatting should be consistent throughout the manuscript. There is a lack of nomenclature to distinguish gene, mRNA from protein levels (eg Ital, etc)

Figure 1 the values for granzyme B are presented as MFI and it's not appropriate to use MFI for a bimodal population - percent positive should be used.

Figure 3. IT is difficult to conclude that ROS produced by MDSC is the cause of reduced mitochondrial function in T cells. Data in Fig 3 is not convincing that T cells aren't making the ROS that affects their function. Presumably the cells have been frozen, cultured, washed etc - how is the ROS still that derived from MDSC. Or is it rather that the ROS from the MDSC reprograms the T cells. How would this work? They also show expression of genes involved in ROS generation and show that there are significantly more in MDSC but they do not show values for T cells. I remain unconvinced that T cells do not make ROS.

For Figure 4 it might be helpful to show bar graphs in log scale. Also the X axis should labeled with what is being measured for clarity. This data shows that the cells are making ROS. It is hard to imagine that the ROS being detected comes from the MDSCs.

Figure 4 E is a bit confusing, since 2 conditions are shown on the left and 3 on the right.

Figure 4F is very interesting and (might) show a selective loss of Complex V. This would suggest less ATP production, but as a consequence could also result in production of more mtROS. The bar graph should focus on individual subunits. Regardless of the mechanism the cells at time point T3 are less metabolically fit but to say they are metabolically paralyzed is a bit of an overstatement. It is not clear in M how glucose uptake is evaluated or calculated.

Figure 5: The data presented aren't sufficient to say that there are mito/TCR complexes (no overlay). More images are needed. Also, is difficult to see differences in the two groups in 5f. More images are needed. Enumerating and quantifying length of mitos here and in Fig7 is challenging. In the images, it is hard to really demarcate individual mitochondria.

Figure 6 it's interesting that they attempt to show that impairing mitochondrial function would result in a similar phenotype. This is a nice set of experiments. It is also pretty well established in the literature that mitochondrial impairment negatively affects T cell function, and it is likely the case here.

Figure 7 provides evidence that improving antioxidant capacity rescues the PITC phenotype. This is nice data. Again the measurements of mitochondrial length and the mitochondrial number is not clear and again the clustering lacks overlay. Also the images in 7H don't look any different.

For the discussion there are some points that need to be toned down and alternative interpretations discussed.

it's not really clear that they've definitively shown that MDSCs increase ROS levels within the CTLs

The cells aren't in metabolic paralysis rather they see a decrease in metabolic fitness.

It is not clear that there is no ROS production in T cells.

Severe mitochondrial dysfunction is an overstatement however there definitely is a reduction and this likely has functional consequences.

The conclusion on line 364 says that the CTL's produce less ATP (What figure does this refer to) and that the T cells failed to localize ATP synthesis to the area of highest demand this bioenergetic failure likely contributes to imperative effector functions - this specific metabolic defects has not been shown.

Referee #1 (Remarks for Author):

This is an interesting study conceptually, and the data overall are quite rigorous (besides the enumeration and measurement of mitochondria, and the data supporting cluster formation). I cannot understand how ROS from MDSC is having the effects they see in culture. I think it is likely that the T cells make the ROS themselves and this could be in part due to loss of complex V. Overall I do think these are quite interesting observations. I do think the mito dysfunction is sufficient to cause the decrease in effector function. And it is possible ROS is involved. But I am not convinced that MDSC are the source of the ROS.

Referee #2 (Comments on Novelty/Model System for Author):

In this study, Simon Hirschberger and colleagues identify a key mechanism driving cytotoxic T cell (CTL) dysfunction following major surgery, which contributes to immune paralysis and heightened risk of nosocomial infections. They show that systemic oxidative stress-mediated by reactive oxygen species (ROS) released from a.o. myeloid-derived suppressor cells (MDSCs)- leads to mitochondrial dysfunction in CTLs. This results in impaired oxidative phosphorylation, disrupted mitochondrial dynamics, and compromised effector function. The use of the mitochondria-targeted antioxidant MitoTEMPO partially restores CTL function, suggesting that preserving mitochondrial integrity could be a viable therapeutic strategy to prevent perioperative immune suppression.

Major Points:

1. Incomplete Timepoint Representation

Several figures only present data for T1 and T2, or T1 and T3, without explaining the omission of a timepoint. This raises concerns about selective reporting. Please either include all timepoints consistently or provide a rationale for each omission.

2. Markers of CTL Exhaustion

The manuscript suggests CTL exhaustion post-surgery based on CD38 expression. While CD38 is relevant-particularly given its role in NAD metabolism and mitochondrial function-it is not sufficient to define exhaustion, especially from a single timepoint. Additional markers (e.g., CD25, CD69 for activation; PD-1, TIM-3 for exhaustion) would significantly strengthen this conclusion.

3. CTL Cytotoxicity Assessment

The impedance-based assay used to assess CTL killing does not directly measure target cell death and can be confounded by factors such as cell proliferation and morphology. Although the authors mention a calcein release assay, corresponding data are not shown and should be included. For greater specificity, a flow cytometry-based killing assay using live/dead staining and a caspase-3 activity reporter is recommended-this would also directly assess granzyme activity and complement Fig. 1G.

4. Blood Composition and CTL Changes Post-Surgery

The manuscript briefly discusses changes in circulating immune cells but requires more detail linking these to PITC. For example, Fig. 1H shows the selective loss of a granzyme B⁺ population rather than a global phenotypic shift-this deserves further discussion. Are these changes due to cellular redistribution or surgery-induced transcriptional reprogramming?

5. Relative ROS Contribution: MDSCs vs. Neutrophils

At T2, neutrophils still comprise ~60% of white blood cells and upregulate NOX2, potentially making them the major source of ROS. While MDSCs express higher NOX2 per cell and increase in frequency, their absolute numbers are still lower. Clarifying the relative contributions of these cell types is critical.

6. Serum Oxidative Stress Impact on CTLs

To support the model linking systemic oxidative stress (Fig. 2) to CTL mitochondrial dysfunction (Fig. 5), the authors should show that CTLs from T1 acquire dysfunction when cultured with T2 serum. This would validate the FCCP-based model and demonstrate functional relevance of serum factors.

7. Control Conditions for MitoTEMPO Rescue Experiments

In Fig. 5 and 6, the rescue of mitochondrial and effector function in T2-CTLs by MitoTEMPO is shown, but T1 and T1+MitoTEMPO controls are missing. These are necessary to demonstrate that the observed effects are specific to oxidative stress and not due to general enhancement of killing capacity.

Minor Points:

1. Metabolic Pathway Analysis

Fig. 2D highlights significant shifts at T3. Please include pathway enrichment data comparing T1 and T3 (as done for T1 vs T2 in Fig. 2E).

2. Figure Clarification

In Fig. 2E, please indicate which dot corresponds to vitamin E metabolism.

3. Figure Readability

Increase the font size and line thickness in Fig. 2F, which is currently unreadable when printed.

4. Proteomic-Metabolomic Correlation

Is there a correlation between proteomic and metabolomic datasets when analyzed in an unbiased, integrative manner? This could add depth to the mechanistic insight.

5. Mitochondrial Mass per Cell

In Fig. 5, include mitochondrial mass per cell as a metric-this is more robust to segmentation artifacts than mitochondrial count per cell (Fig. 5G).

6. ROS Source Validation

To directly compare the ROS output of neutrophils and MDSCs, the authors should consider using a luminol-based assay on sorted populations from T1 and T2 samples. This would help resolve the key question of ROS source post-surgery.

Referee #3 (Comments on Novelty/Model System for Author):

The authors employ a comprehensive array of omics approaches together with in vitro studies to identify novel regulators of metabolic paralysis following major surgeries. Elucidating the mechanisms underlying this immune dysfunction is of high clinical relevance, as it has the potential to benefit a large patient population affected by post-surgical immune paralysis, including the prevention of complications such as nosocomial infections. The methodological strategy-integrating patient-derived data with mechanistic in vitro experiments-is well-conceived and highly appropriate for uncovering previously unrecognized determinants of immune suppression in this context.

Referee #3 (Remarks for Author):

In this study, Hirschberger et al. investigate the immune paralysis that emerges following major surgery by analyzing changes in cytotoxic CD8+ T cells (CTLs) at distinct time points in patients undergoing surgical procedures. The authors identify immunological alterations in CTLs after surgery, including upregulation of the exhaustion marker CD38, along with reduced cytokine secretion and degranulation, ultimately leading to diminished cytotoxic activity. To uncover the underlying mechanisms, the authors profiled the serum proteome and metabolome of patients. This approach revealed an oxidative signature, which they attribute to the increased presence of myeloid-derived suppressor cells (MDSCs). The authors propose that the oxidative signature of MDSCs induces bystander oxidative stress in CTLs, leading to their metabolic failure marked by mitochondrial defects, mitochondrial hyperfusion, and impaired immune synapse formation, thereby contributing to CTL dysfunction-paralysis found in post-surgical patients.

This study is of great interest, as it uncovers both immune and metabolic changes in CTLs that may underpin the observed immune paralysis in patients undergoing major surgery. It highlights the relevance of immunometabolic regulation in this clinical context and proposes a potential mechanism through which MDSCs may impair CTL function via oxidative stress, opening therapeutic opportunities to modulate or restore immune paralysis in patients undergoing major surgeries.

While the methodology followed in this study is very compelling, several aspects of the study would benefit from further experimental validation to strengthen the conclusions and mechanistic interpretation raised by the authors.

This is specially relevant for the role of MDSCs in mediating the oxidative stress found in CTLs.

The current data on the study remain correlative regarding the role of MDSCs in driving CTL metabolic failure. To substantiate this central claim, the authors should perform, for instance, co-culture experiments using MDSCs isolated from patients after surgery and CTLs from healthy donors to test whether MDSCs post-surgery induce oxidative stress and mitochondrial defects in CTLs. Moreover, authors should assess whether ROS scavengers such as MitoTempo can rescue CTL mitochondrial and functional defects in these co-culture settings. These proposed approaches or similar would strengthen the authors' conclusions on the role of oxidative stress in these settings, and help to provide a causative role of MDSCs in CTL paralysis.

Other minor comments

- Direct evidence of increased ROS in MDSCs is currently lacking. The authors should quantify cellular and/or mitochondrial ROS in MDSCs (e.g., CellROX, MitoSOX).
- The authors should include additional surface markers of activation and exhaustion on CTLs from patients to provide a more comprehensive understanding of their immune status post-surgery (e.g., PD-1, TIM-3, LAG-3, CD69, CD25).
- Please revise the X-axis labels in the FACS histograms shown in Figures 4 and 5a,b to include the fluorophore or dye used, as shown in other panels (e.g., Figure 1d). This will improve clarity and consistency.
- How immune synapse formation assays were performed requires clarification. How were CTLs stimulated (e.g., anti-CD3/CD28 beads) and for how long? Was immune synapse formation assessed while CTLs contacted the beads? Please clarify this point.

Mitochondrial damage drives T-cell immunometabolic paralysis after major surgery

Point by point response to the reviewer's comments

Referee #1 (Comments on Novelty/Model System for Author):

Hirschberger et al propose that following surgery T cells undergo metabolic dysfunction they propose that ROS derived from myeloid suppressor cells impaired mitochondrial function in T cells and influence T cell activity. They call this immune paralysis. Overall this work does offer new insights into mechanisms driving immune dysfunction following surgery. They suggest that ROS levels are contributing to T cell dysfunction and that T cell function may be partially restored by antioxidant activity. Overall, the manuscript is well written and the data are rigorous. There are several revisions that would improve the clarity and quality of the manuscript. Further there are alternate interpretations of their data that should be considered.

Critique:

1. General: There is lack of consistency in figures with flow histograms accompanied by bar/scatter graphs. Sometimes left to right in the bar graph matches top to bottom (Fig 4 a-d) and sometimes it is the opposite (Fig 1d,h, 4e). Formatting should be consistent throughout the manuscript. There is a lack of nomenclature to distinguish gene, mRNA from protein levels (eg Ital, etc).

#1 We thank the referee for his remark. We have changed the flow histograms to provide consistency throughout the Figures. Also, we have checked the formatting throughout the manuscript to enable clear discrimination between mRNA and protein levels.

2. Figure 1 the values for granzyme B are presented as MFI and it's not appropriate to use MFI for a bimodal population - percent positive should be used.

#2 We have changed the values for GZMB and PRF1 to account for the bimodal populations as requested (Fig 1 H/I).

3. Figure 3. It is difficult to conclude that ROS produced by MDSC is the cause of reduced mitochondrial function in T cells. Data in Fig 3 is not convincing that T cells aren't making the ROS that affects their function. Presumably the cells have been frozen, cultured, washed etc - how is the ROS still that derived from MDSC. Or is it rather that the ROS from the MDSC reprograms the T cells. How would this work? They also show expression of genes involved in ROS generation and show that there are significantly more in MDSC but they do not show values for T cells. I remain unconvinced that T cells do not make ROS.

#3 We thank the reviewer for the insightful comments on this central point of our manuscript. We carefully considered all suggestions regarding the source of ROS (points 3., 8., 10., and Remarks for Author) and performed additional experiments to further investigate the potential contribution of a CTL-intrinsic source of ROS to mitochondrial dysfunction: Newly performed co-culture experiments demonstrate that postoperative MDSC – not PMN – increase mitochondrial ROS and impair mitochondrial mass and membrane potential in

healthy CTL (new Fig. EV4B–D, results, page 8). Additionally, serum-transfer experiments using postoperative patient serum elicited a similar CTL phenotype (new Fig. EV4E–G, results, page 9), indicating that MDSC-derived factors—most likely ROS or ROS-producing enzymes—initiate mitochondrial damage in CTL. This damage subsequently triggers a cell-intrinsic amplification of mitochondrial ROS production.

Supporting this model, we observed reduced expression of ETC complexes in CTL after major surgery, with the most pronounced effects in complex V and a significant decrease in complex I (new sentence in results, page 8 and revised graph in Fig. 4), both of which have been linked to dysregulated mitochondrial ROS (Okoye et al, 2023).

Taken together, we propose a two-step mechanism underlying postoperative CTL suppression: (i) primary mitochondrial damage caused by MDSC-derived factors such as ROS and ROS-generating enzymes, and (ii) secondary ROS amplification within CTL driven by mitochondrial dysfunction.

These new data and their mechanistic implications now are discussed on page x, lines y and have also been integrated into the new summary.

Regarding the additionally requested points:

The cells analyzed in Figure 3 were isolated and lysed immediately after blood withdrawal, without freezing or culturing. Gene expression analysis of ROS-producing enzymes in T cells (Fig. 3A/B) demonstrates minimal to no expression in CTL.

4. For Figure 4 it might be helpful to show bar graphs in log scale. Also the X axis should be labeled with what is being measured for clarity. This data shows that the cells are making ROS. It is hard to imagine that the ROS being detected comes from the MDSCs.

Figure 4 E is a bit confusing, since 2 conditions are shown on the left and 3 on the right. Figure 4F is very interesting and (might) show a selective loss of Complex V. This would suggest less ATP production, but as a consequence could also result in production of more mtROS. The bar graph should focus on individual subunits. Regardless of the mechanism the cells at time point T3 are less metabolically fit but to say they are metabolically paralyzed is a bit of an overstatement. It is not clear how glucose uptake is evaluated or calculated.

#4 The labeling of the axis has been improved to clearly show what is being measured. We added the missing time point to Figure 4e. We have now added a bar graph depicting densitometric quantification of individual complexes as requested (Fig 4g). The implications of these findings are discussed in response #3.

We have also toned down the term “paralyzed” and we have added the missing description of cellular glucose uptake to the materials and methods section.

5. Figure 5: The data presented aren't sufficient to say that there are mito/TCR complexes (no overlay). More images are needed. Also, it is difficult to see differences in the two groups in 5f. More images are needed. Enumerating and quantifying length of mitos here and in Fig7 is challenging. In the images, it is hard to really demarcate individual mitochondria.

#5 The term mito/TCR complex is indeed misleading, we apologize for this. Our confocal microscopy data is not intended to propose an overlay of TCR and mitochondria, which has also not been described yet. Mitochondrial translocation towards the T-cell Immunological Synapse (IS) is characterized by an assembling of most mitochondrial structures near one site of the plasma membrane together with a close proximity of TCR and Mitochondria, that

has been described as a distance $<1\mu\text{m}$ (<https://doi.org/10.1073/pnas.0703126104>). We agree that the word “complex” might be misleading in this context and have rephrased the respective parts of the manuscript. This approximation of TCR/plasma membrane (as indicated by anti-CD3) and mitochondria (as indicated by MitoTracker) can now be seen in our new exemplary pictures (Fig 5 I, Fig 7 H).

To show a better visualization of mitochondrial length evaluation we have now also included skeleton images of MitoTracker staining processed with the Mitochondrial Analyzer ImageJ plugin, as previously described (Chaudhry et al, 2020) and validated (Hemel et al, 2021). These analyses - also including new “mean mitochondrial area” (measured automatically by the software) to supplement the manually performed measurement of mitochondrial length - have now been added to Figure 5.

6. Figure 6 it's interesting that they attempt to show that impairing mitochondrial function would result in a similar phenotype. This is a nice set of experiments. It is also pretty well established in the literature that mitochondrial impairment negatively affects T cell function, and it is likely the case here.

7. Figure 7 provides evidence that improving antioxidant capacity rescues the PITC phenotype. This is nice data. Again the measurements of mitochondrial length and the mitochondrial number is not clear and again the clustering lacks overlay. Also the images in 7H don't look any different.

#7 As stated in response #5, we have clarified the measurement of mitochondrial length and number, and we have added new exemplary pictures (Fig 7 H).

8. For the discussion there are some points that need to be toned down and alternative interpretations discussed.

It's not really clear that they've definitively shown that MDSCs increase ROS levels within the CTLs.

#8 As stated in response #3, we have implemented new experiments and have revised the results and discussion section accordingly.

9. The cells aren't in metabolic paralysis rather they see a decrease in metabolic fitness.

#9 Within the manuscript, we now omitted the term “metabolic paralysis and changed it to “decrease of immunometabolic fitness” (results, page 8+9, discussion, page 11).

10. It is not clear that there is no ROS production in T cells.

#10 This appears to be a misunderstanding. We did not claim that CTL do not produce ROS, but that the strong post-surgery increase of ROS in CTL cannot be explained by ROS production through classical cellular enzymes. In new experiments, we now demonstrate that MDSC-derived ROS damage CTL mitochondria, causing electron leakage and secondary ROS amplification. Thus, the post-surgery ROS elevation likely results from (i) primary MDSC-induced damage and (ii) secondary mitochondrial dysfunction within CTL. (please see response #3). These findings have been incorporated into the revised results and discussion (page 8).

11. Severe mitochondrial dysfunction is an overstatement however there definitely is a reduction and this likely has functional consequences.

#11 We now toned down the term as requested (results, page 8+9).

12. The conclusion on line 364 says that the CTL's produce less ATP (What figure does this refer to) and that the T cells failed to localize ATP synthesis to the area of highest demand this bioenergetic failure likely contributes to imperative effector functions -this specific metabolic defects has not been shown.

#12 We agree with the reviewer that -although oxygen consumption is a surrogate parameter for ATP synthesis- the Seahorse data do not directly depict ATP synthesis. We thus replaced the term "ATP synthesis" with "mitochondrial oxidative phosphorylation".

We also agree with the reviewer that our data does not provide intracellular spatial resolution of the OXPHOS function. We have now indicated the speculative character of our conclusion (discussion page 12+13).

Referee #1 (Remarks for Author):

This is an interesting study conceptually, and the data overall are quite rigorous (besides the enumeration and measurement of mitochondria, and the data supporting cluster formation). I cannot understand how ROS from MDSC is having the effects they see in culture. I think it is likely that the T cells make the ROS themselves and this could be in part due to loss of complex V. Overall I do think these are quite interesting observations. I do think the mito dysfunction is sufficient to cause the decrease in effector function. And it is possible ROS is involved. But I am not convinced that MDSC are the source of the ROS.

We thank the reviewer for this valuable feedback. As stated in response #3, we have now designed and performed various new experiments, supporting the reviewer's view, which has been integrated and added to the manuscript as described above.

Referee #2 (Comments on Novelty/Model System for Author):

In this study, Simon Hirschberger and colleagues identify a key mechanism driving cytotoxic T cell (CTL) dysfunction following major surgery, which contributes to immune paralysis and heightened risk of nosocomial infections. They show that systemic oxidative stress-mediated by reactive oxygen species (ROS) released from a.o. myeloid-derived suppressor cells (MDSCs)-leads to mitochondrial dysfunction in CTLs. This results in impaired oxidative phosphorylation, disrupted mitochondrial dynamics, and compromised effector function. The use of the mitochondria-targeted antioxidant MitoTEMPO partially restores CTL function, suggesting that preserving mitochondrial integrity could be a viable therapeutic strategy to prevent perioperative immune suppression.

Major Points:

1. Incomplete Timepoint Representation

Several figures only present data for T1 and T2, or T1 and T3, without explaining the omission of a timepoint. This raises concerns about selective reporting. Please either include all timepoints consistently or provide a rationale for each omission.

#1 Selective reporting was not performed. In Figure 1F, the degranulation assay was omitted at time point T3 to ensure methodological comparability. This assay involves a complex experimental protocol that is particularly sensitive to timing deviations during the multiple incubation and staining steps. Samples from T1 and T2 were generated on the same day and could therefore be processed simultaneously on the following day, a requirement that could not be fulfilled for T3.

Proteomic and metabolomic analyses were conducted at all three time points (T1–T3, Fig. 2B, D, E, F, G and Fig. EV2). We subsequently assessed mitochondrial damage in CTL and identified the most pronounced phenotype at T2. As a result, downstream analyses focused on comparing T1 and T2 /Fig. 4C). This clarification has been added to the manuscript (Results, page 9).

2. Markers of CTL Exhaustion

The manuscript suggests CTL exhaustion post-surgery based on CD38 expression. While CD38 is relevant-particularly given its role in NAD metabolism and mitochondrial function-it is not sufficient to define exhaustion, especially from a single timepoint. Additional markers (e.g., CD25, CD69 for activation; PD-1, TIM-3 for exhaustion) would significantly strengthen this conclusion.

#2 Thank you for this valuable suggestion. As suggested, we performed additional experiments applying a more comprehensive T cell activation and exhaustion panel using the proposed cell markers and added these results to our manuscript (Fig EV 1 A + B). In line with the fact that classical exhaustion markers are known for their delayed upregulation, we did not detect any significant changes. To represent this accurately, we refrained from using the wording “exhausted phenotype and changed it to “dysfunctional” (results, page 5).

3. CTL Cytotoxicity Assessment

The impedance-based assay used to assess CTL killing does not directly measure target cell death and can be confounded by factors such as cell proliferation and morphology. Although the authors mention a calcein release assay, corresponding data are not shown and should be included. For greater specificity, a flow cytometry-based killing assay using live/dead staining and a caspase-3 activity reporter is recommended-this would also directly assess granzyme activity and complement Fig. 1G.

#3 Thank you for this valuable remark. We now enrolled additional patients and performed a flow cytometry-based cytotoxicity assay using live-dead staining, which has been added to the manuscript (Fig EV 1 C). Additionally, we also performed a Caspase 3/7 activity assay, to account for direct assessment of granzyme activity (Fig EV 1 D). This data corroborates our initial findings and is now included in the results and materials/methods section of our manuscript (results, page 5, materials and methods, page 20).

4. Blood Composition and CTL Changes Post-Surgery

The manuscript briefly discusses changes in circulating immune cells but requires more detail linking these to PITC. For example, Fig. 1H shows the selective loss of a granzyme B⁺

population rather than a global phenotypic shift-this deserves further discussion. Are these changes due to cellular redistribution or surgery-induced transcriptional reprogramming?

#4 This impression might have been evoked by the misleading arrangement of the time points in the flow cytometry histograms between the figures. We apologize and have corrected these histograms. After major surgery, we detect a significant increase of GZMB+ CTL. We further link this finding - in synopsis with abrogated GZMB secretion of these cells - to a compromised exocytosis of cytolytic granules (Fig 1g-j).

5. Relative ROS Contribution: MDSCs vs. Neutrophils

At T2, neutrophils still comprise ~60% of white blood cells and upregulate NOX2, potentially making them the major source of ROS. While MDSCs express higher NOX2 per cell and increase in frequency, their absolute numbers are still lower. Clarifying the relative contributions of these cell types is critical.

#5 At T1, no MDSC were detectable. After surgery at T2, MDSC emerged, comprising between 15% and 50% of PBMC ((Hübner et al, 2019). Neutrophils showed no change in MPO expression between T1 and T2, with only a slight, non-significant increase in NOX2. In contrast, MDSC expressed MPO at levels 45-fold higher and NOX2 at 3.5-fold higher than neutrophils. Supporting this, additional experiments using a luminol-based ROS detection assay with sorted MDSC and T2 neutrophils revealed significantly greater chemiluminescence in MDSC (Fig. EV3, results, page 7+8).

We performed additional co-culture experiments using T1 CTL cultured with either T1 or T2 neutrophils, or with MDSC. Only MDSC induced significant mitochondrial damage (Fig. EV4B-E).

6. Serum Oxidative Stress Impact on CTLs

To support the model linking systemic oxidative stress (Fig. 2) to CTL mitochondrial dysfunction (Fig. 5), the authors should show that CTLs from T1 acquire dysfunction when cultured with T2 serum. This would validate the FCCP-based model and demonstrate functional relevance of serum factors.

#6 We evaluated this valuable suggestion in additional experiments and were indeed able to show that T1 CTL function and metabolism is impaired after incubation with T2 serum compared to incubations conducted with T1 serum. These results are now depicted in Fig EV4 E-G, and have been added to the results and discussion of the manuscript (results, page 9, discussion, page 13).

7. Control Conditions for MitoTEMPO Rescue Experiments

In Fig. 5 and 6, the rescue of mitochondrial and effector function in T2-CTLs by MitoTEMPO is shown, but T1 and T1+MitoTEMPO controls are missing. These are necessary to demonstrate that the observed effects are specific to oxidative stress and not due to general enhancement of killing capacity.

#7 As requested, we have now implemented the missing controls T1/T1+MT (including OXPHOS function, cytotoxicity and mitochondrial mass), showing no effect of MT on healthy CTL (Fig 7, Fig EV5).

Minor Points:

1. Metabolic Pathway Analysis

Fig. 2D highlights significant shifts at T3. Please include pathway enrichment data comparing T1 and T3 (as done for T1 vs T2 in Fig. 2E).

#1 We have now included pathway enrichment data T1 vs T3 as requested (Fig 2G).

2. Figure Clarification

In Fig. 2E, please indicate which dot corresponds to vitamin E metabolism.

#2 We have corrected the Figure and now show the dot for vitamin E metabolism (Fig 2 E).

3. Figure Readability

Increase the font size and line thickness in Fig. 2F, which is currently unreadable when printed.

#3 We have changed the figure as suggested to optimize readability and thank the reviewer for this important remark (Fig 2 H).

4. Proteomic-Metabolomic Correlation

Is there a correlation between proteomic and metabolomic datasets when analyzed in an unbiased, integrative manner? This could add depth to the mechanistic insight.

#4 Since proteomic and metabolomic data derive from different patient samples, data-driven omics integration was not suitable. However, we now have added a knowledge-driven joint pathway analysis (Fig. EV2 D).

5. Mitochondrial Mass per Cell

In Fig. 5, include mitochondrial mass per cell as a metric-this is more robust to segmentation artifacts than mitochondrial count per cell (Fig. 5G).

#5 We have assessed mitochondrial mass using flow cytometry (MitoTracker). To account for the reviewers' concerns, we have additionally performed analysis of the mitochondrial network using Mitochondrial Analyzer ImageJ plugin as previously described (Chaudhry et al, 2020) and validated (Hemel et al, 2021). These analyses - also including new "mean mitochondrial area per cell" have now been added to Figure 5.

6. ROS Source Validation

To directly compare the ROS output of neutrophils and MDSCs, the authors should consider using a luminol-based assay on sorted populations from T1 and T2 samples. This would help resolve the key question of ROS source post-surgery.

#6 We thank the reviewer for this important suggestion. We now have applied a luminol-based assay on sorted MDSC and neutrophils, indicating significantly higher ROS production by MDSC (Fig. EV3, results, page 7+8).

Referee #3 (Comments on Novelty/Model System for Author):

The authors employ a comprehensive array of omics approaches together with in vitro studies to identify novel regulators of metabolic paralysis following major surgeries. Elucidating the mechanisms underlying this immune dysfunction is of high clinical relevance, as it has the potential to benefit a large patient population affected by post-surgical immune paralysis, including the prevention of complications such as nosocomial infections. The methodological strategy-integrating patient-derived data with mechanistic in vitro experiments-is well-

conceived and highly appropriate for uncovering previously unrecognized determinants of immune suppression in this context.

Referee #3 (Remarks for Author):

In this study, Hirschberger et al. investigate the immune paralysis that emerges following major surgery by analyzing changes in cytotoxic CD8+ T cells (CTLs) at distinct time points in patients undergoing surgical procedures. The authors identify immunological alterations in CTLs after surgery, including upregulation of the exhaustion marker CD38, along with reduced cytokine secretion and degranulation, ultimately leading to diminished cytotoxic activity. To uncover the underlying mechanisms, the authors profiled the serum proteome and metabolome of patients. This approach revealed an oxidative signature, which they attribute to the increased presence of myeloid-derived suppressor cells (MDSCs). The authors propose that the oxidative signature of MDSCs induces bystander oxidative stress in CTLs, leading to their metabolic failure marked by mitochondrial defects, mitochondrial hyperfusion, and impaired immune synapse formation, thereby contributing to CTL dysfunction-paralysis found in post-surgical patients.

This study is of great interest, as it uncovers both immune and metabolic changes in CTLs that may underpin the observed immune paralysis in patients undergoing major surgery. It highlights the relevance of immunometabolic regulation in this clinical context and proposes a potential mechanism through which MDSCs may impair CTL function via oxidative stress, opening therapeutic opportunities to modulate or restore immune paralysis in patients undergoing major surgeries.

While the methodology followed in this study is very compelling, several aspects of the study would benefit from further experimental validation to strengthen the conclusions and mechanistic interpretation raised by the authors.

This is especially relevant for the role of MDSCs in mediating the oxidative stress found in CTLs.

The current data on the study remain correlative regarding the role of MDSCs in driving CTL metabolic failure. To substantiate this central claim, the authors should perform, for instance, co-culture experiments using MDSCs isolated from patients after surgery and CTLs from healthy donors to test whether MDSCs post-surgery induce oxidative stress and mitochondrial defects in CTLs. Moreover, authors should assess whether ROS scavengers such as MitoTempo can rescue CTL mitochondrial and functional defects in these co-culture settings. These proposed approaches or similars would strengthen the authors' conclusions on the role of oxidative stress in these settings and help to provide a causative role of MDSCs in CTL paralysis.

#1 We thank the reviewer for this valuable suggestion. As recommended, we have performed co-incubation experiments using MDSC from T2 and incubated them with T1 CTL from the same patient. This led to an increase of mitochondrial ROS in these CTL, a decline of mitochondrial membrane potential and a profound abrogation of mitochondrial mass, corroborating the role of MDSC. MitoTEMPO treatment of these T1 CTL co-cultivated with MDSC was able to ameliorate this mitochondrial damage. We have implemented this data into the manuscript (new Fig. EV4B–D, EV 5 E, results, page 8+10, discussion page 13).

Other minor comments

- Direct evidence of increased ROS in MDSCs is currently lacking. The authors should quantify cellular and/or mitochondrial ROS in MDSCs (e.g., CellROX, MitoSOX).

#2 A comparison of ROS levels in MDSCs before and after surgery is not feasible, as MDSCs are not present at time point T1. MDSC constitutively express high levels of ROS-producing enzymes and predominantly release ROS into the extracellular space, thereby limiting detection by intracellular probes such as CellROX or MitoSOX as evidenced by new data (Fig EV4 A).

To demonstrate ROS production by MDSCs, we now have performed a luminol-based assay on sorted MDSCs and T2 neutrophils, measuring extracellularly released ROS. This revealed markedly higher ROS production by MDSCs (results Fig. EV3, results, page 7+8).

- The authors should include additional surface markers of activation and exhaustion on CTLs from patients to provide a more comprehensive understanding of their immune status post-surgery (e.g., PD-1, TIM-3, LAG-3, CD69, CD25).

#3 We have now implemented the requested experiments on the surface markers as requested (results page 5, Fig EV1B and Fig 1F).

- Please revise the X-axis labels in the FACS histograms shown in Figures 4 and 5a,b to include the fluorophore or dye used, as shown in other panels (e.g., Figure 1d). This will improve clarity and consistency.

#4 We have revised the respective histograms as suggested.

- How immune synapse formation assays were performed requires clarification. How were CTLs stimulated (e.g., anti-CD3/CD28 beads) and for how long? Was immune synapse formation assessed while CTLs contacted the beads? Please clarify this point.

#5 Thank you, we apologize for the missing information. CTL were stimulated using anti-CD3/CD28 beads for 20h. Immediately prior to fixation and preparation for microscopy, the dynabeads were magnetically removed to avoid fluorescence contamination. We have now included these details to the methods section.

References:

Chaudhry A, Shi R & Luciani DS (2020) A pipeline for multidimensional confocal analysis of mitochondrial morphology, function, and dynamics in pancreatic β -cells. *Am J Physiol Endocrinol Metab* 318: E87–E101

Hemel IMG, Engelen BPH, Lubber N & Gerards M (2021) A hitchhiker's guide to mitochondrial quantification. *Mitochondrion* 59: 216–224

Hübner M, Tomasi R, Effinger D, Wu T, Klein G, Bender M, Kilger E, Juchem G, Schwedhelm E & Kreth S (2019) Myeloid-Derived Suppressor Cells Mediate Immunosuppression After Cardiopulmonary Bypass. *Crit Care Med* 47: e700–e709

Okoye CN, Koren SA & Wojtovich AP (2023) Mitochondrial complex I ROS production and redox signaling in hypoxia. *Redox Biol* 67: 102926

17th Sep 2025

Dear Prof. Kreth,

Thank you for the submission of your revised manuscript to EMBO Molecular Medicine. I am pleased to inform you that we will be able to accept your manuscript pending the following final amendments:

- 1) Please implement referee #1 and #3 suggestions.
- 2) Figures:
 - We note that some panels are reused. Figure 3C T2 FACS plot is reused in Figure EV3A and Figure 5I right top image is reused in Figure 7H. Please cite in the respective figure legend every reused image/panel.
 - Please label staining/colors of the microscopic images in the figures.
- 3) In the main manuscript file, please do the following:
 - Please address all comments suggested by our data editors listed below:
 - o Data availability statement:
 1. Please note that the specific URLs for E-MTAB-14974, MSV000097035, S-BIAD2197 datasets are not provided in the data availability statement.
 - o Figure legends:
 1. Please note that the exact p values are not provided in the legends of figures 1B-I; 2H, 3A, B, C, G H; 4A, B, C, D, E, G, I, J, K, N; 5A-C, E, G, H, I; 6A, D, E, G, H, I, J; 7A, D, E, F, G, J; EV1 B, EV2 B, EV5 E.
 2. Please note that the box plots need to be defined in terms of minima, maxima, centre, bounds of box and whiskers, and percentile in the legends of figures 1F, 2H; 4B, E, I-L, N; 5A-C, G, I; 7F, G, J; EV2 B.
 3. Please note that information related to n is missing in the legend of figure 3C
 - Add up to 5 keywords.
 - Correct callouts of Supplementary Table 1 to Table EV1. Please make sure that all callouts are correct.
 - Please remove One Sentence Summary, Code Availability, Additional Information and Lead Contact sections.
 - Remove BioRender information from Acknowledgements and add the following paragraph in Methods:

Graphics:

(some of the... OR Figure #... OR synopsis) Graphics were created with BioRender.com.

- In Methods, provide the statement that in addition to informed consent and WMA Declaration of Helsinki the experiments also conformed to the principles set out in the and the Department of Health and Human Services Belmont Report.
- Please remove Reagents and Tools Table from the manuscript file and uploaded it as a separate file. More information on how to adhere to this format as well as downloadable templates (.docx) for the Reagents and Tools Table can be found in our author guidelines: <https://www.embopress.org/page/journal/17574684/authorguide#structuredmethods>

An example of a paper with Structured Methods can be found here:

<https://www.embopress.org/doi/full/10.1038/s44320-024-00037-6#sec-4>

- Rename "Competing interests statement" to "Disclosure and competing interests statement". We updated our journal's competing interests policy in January 2022 and request authors to consider both actual and perceived competing interests. Please review the policy <https://www.embopress.org/competing-interests> and update your competing interests if necessary.
- Author contributions: Please remove it from the manuscript and specify author contributions in our submission system. CRediT has replaced the traditional author contributions section because it offers a systematic machine-readable author contributions format that allows for more effective research assessment. You are encouraged to use the free text boxes beneath each contributing author's name to add specific details on the author's contribution. More information is available in our guide to authors:

<https://www.embopress.org/page/journal/17574684/authorguide#authorshipguidelines>

- Please use the following format to report the accession number of your data:

[data type]: [full name of the resource] [accession number/identifier] ([doi or URL or identifiers.org/DATABASE:ACCESSION])

Please check "Author Guidelines" for more information.

<https://www.embopress.org/page/journal/17574684/authorguide#availabilityofpublishedmaterial>.

- 4) Source data: Please upload the source data files as one (ZIP) file per figure and upload a completed source data checklist sent to you on June 16.
- 5) Funding: Please merge it with Acknowledgments.
- 6) The Paper Explained: Please add it to the main manuscript text.
- 7) Synopsis:
 - Synopsis image: Please resize the image to 550 px-wide x 300-600 pixels high and upload it as a high-resolution jpeg file.
 - Please check your synopsis text and image before submission with your revised manuscript. Please be aware that in the proof stage minor corrections only are allowed (e.g., typos).
- 8) As part of the EMBO Publications transparent editorial process initiative (see our Editorial at

<http://embomolmed.embopress.org/content/2/9/329>), EMBO Molecular Medicine will publish online a Review Process File (RPF) to accompany accepted manuscripts. This file will be published in conjunction with your paper and will include the anonymous referee reports, your point-by-point response and all pertinent correspondence relating to the manuscript. Let us know whether you agree with the publication of the RPF and as here, if you want to remove or not any figures from it prior to publication. Please note that the Authors checklist will be published at the end of the RPF.

9) Please provide a point-by-point letter INCLUDING my comments as well as the reviewer's reports and your detailed responses (as Word file).

I look forward to reading a new revised version of your manuscript as soon as possible.

Yours sincerely,

Zeljko Durdevic

Zeljko Durdevic
Senior Editor
EMBO Molecular Medicine

*** Instructions to submit your revised manuscript ***

1) a .docx formatted version of the manuscript text (including Figure legends and tables)

2) Separate figure files*

3) supplemental information as Expanded View and/or Appendix. Please carefully check the authors guidelines for formatting Expanded view and Appendix figures and tables at <https://www.embopress.org/page/journal/17574684/authorguide#expandedview>

4) a letter INCLUDING the reviewer's reports and your detailed responses to their comments (as Word file).

5) The paper explained: EMBO Molecular Medicine articles are accompanied by a summary of the articles to emphasize the major findings in the paper and their medical implications for the non-specialist reader. Please provide a draft summary of your article highlighting

6) Author contributions: the contribution of every author must be detailed in a separate section.

7) EMBO Molecular Medicine now requires a complete author checklist (<https://www.embopress.org/page/journal/17574684/authorguide>) to be submitted with all revised manuscripts. Please use the

checklist as guideline for the sort of information we need WITHIN the manuscript. The checklist should only be filled with page numbers where the information can be found. This is particularly important for animal reporting, antibody dilutions (missing) and exact values and n that should be indicated instead of a range.

8) Every published paper now includes a 'Synopsis' to further enhance discoverability. Synopses are displayed on the journal webpage and are freely accessible to all readers. They include a short stand first (maximum of 300 characters, including space) as well as 2-5 one sentence bullet points that summarise the paper. Please write the bullet points to summarise the key NEW findings. They should be designed to be complementary to the abstract - i.e. not repeat the same text. We encourage inclusion of key acronyms and quantitative information (maximum of 30 words / bullet point). Please use the passive voice. Please attach these in a separate file or send them by email, we will incorporate them accordingly.

You are also welcome to suggest a striking image or visual abstract to illustrate your article. If you do please provide a jpeg file 550 px-wide x 300-600px high.

9) A Conflict of Interest statement should be provided in the main text

10) Please note that we now mandate that all corresponding authors list an ORCID digital identifier. This takes <90 seconds to complete. We encourage all authors to supply an ORCID identifier, which will be linked to their name for unambiguous name identification.

Currently, our records indicate that the ORCID for your account is 0000-0002-0073-2098.

Link Not Available

11) Include a Reagents and Tools Table as part of the Methods section, which can be downloaded from our author guidelines (<https://www.embopress.org/page/journal/17574684/authorguide#structuredmethods>)

Photos 400-800 DPI

*Additional important information regarding figures and illustrations can be found at <https://bit.ly/EMBOPressFigurePreparationGuideline>. See also figure legend preparation guidelines: <https://www.embopress.org/page/journal/17574684/authorguide#figureformat>

***** Reviewer's comments *****

Referee #1 (Comments on Novelty/Model System for Author):

Overall the manuscript has been significantly improved.

Referee #1 (Remarks for Author):

Overall the manuscript has been significantly improved. The authors responded well to all of the reviewers comments. Some minor points should be addressed.

Minor points

P8, discussion of Fig EV4 (Figure # on the figures would have been helpful) I think there is at type of "distorted" mitochondrial mass instead of "decreased" mitochondrial mass.

Description of fig 2E/G can more pathways be labeled? This is interesting data and it is not expected that the group follows up on all of the pathways - but it would be good to present them.

Figure EV2D could go in main figures if it fits well.

Fig 3A if they have NOX2 ICS that would be great, but if not it is not required.

Can anything be added/annotated to Fig D/E to illustrate something other than differentially expressed genes. OR swapped with new data presented in EV4.

P 9" marked by impaired oxidative phosphorylation" = decreased resting OCR, coupled respiration and maximal respiration. Do they have the data for activity per mito to present the ratio. I.e TMRM/mitotracker/spy. It seems that there is perhaps the same mito activity per mitomass, just lower mito mass.

The new data does not all need to go in supplements. Consider putting all key findings in main figures.

Referee #2 (Remarks for Author):

Is suitable for publication

Referee #3 (Comments on Novelty/Model System for Author):

The study uses an appropriate and technically sound approach, combining patient samples with multiple omics platforms to obtain a comprehensive characterization, followed by in vitro assays that provide mechanistic insights. The integration of clinical material with experimental validation enhances the robustness of the findings and supports the adequacy of the model. However, the presentation of some data could be improved for clarity. For instance, volcano plots would benefit from including metabolite or enzyme names and indicating the omics technique used (e.g., proteomics, metabolomics) to facilitate interpretation. Similarly, the the analysis of the RNA-seq dataset would be strengthened by a more systematic exploration of differentially regulated pathways and processes, beyond the focus on oxidative stress-related genes. These findings have clear medical relevance, as they provide insights into how postoperative immune suppression may be metabolically mediated. Understanding this mechanism opens potential avenues for therapeutic interventions to preserve T cell function in the postsurgical setting, which could have direct clinical impact on patient outcomes. As the authors have performed and included the additional experiments suggested in my initial review, I consider that the manuscript is now suitable for publication pending minor corrections. Most of these relate to editing and presentation. In particular, figure panels, labels, and graphs are generally too small, which makes interpretation difficult, and the extensive use of acronyms further complicates figure readability. Addressing these issues will substantially improve the accessibility of the manuscript.

Referee #3 (Remarks for Author):

Thank you for taking into consideration my suggestions and comments. I only have a few remaining points for clarification and minor corrections:

Figure 1: Please clarify whether the parameters shown were measured in freshly isolated CTLs or in cells restimulated (e.g., with CD3/CD28 beads and/or PMA/ionomycin). This information is essential for data interpretation.

Page 36, Figure 4f: The exemplary immunoblot of OXPHOS complexes in CTLs should be described more precisely. Please indicate that the blot shows representative proteins from the different OXPHOS complexes, rather than the intact complexes themselves. Ideally, specify in the figure legend which exact protein(s) were detected. Please, also correct this along the manuscript.

Figure EV4F: The sentence should be corrected from "compromised membrane integrity" to "compromised mitochondrial membrane potential integrity" for accuracy.

Figure EV4G: The data on impaired mitochondrial respiratory function are shown without the number of patients analyzed or a quantification. Please include both to strengthen this figure.

Referee #1 (Remarks for Author):

Overall the manuscript has been significantly improved. The authors responded well to all of the reviewers comments. Some minor points should be addressed.

Minor points:

P8, discussion of Fig EV4 (Figure # on the figures would have been helpful) I think there is at type of "distorted" mitochondrial mass instead of "decreased" mitochondrial mass.

We changed the term "decreased" to "distorted" as requested.

Description of fig 2E/G can more pathways be labeled? This is interesting data and it is not expected that the group follows up on all of the pathways - but it would be good to present them.

We now doubled the labeled pathways of Fig 2E/G each.

Figure EV2D could go in main figures if it fits well.

We have moved Fi EV2D to the main figure (Fig 2 H).

Fig 3A if they have NOX2 ICS that would be great, but if not it is not required.

We apologize as we have not performed NOX2 ICS.

Can anything be added/annotated to Fig D/E to illustrate something other than differentially expressed genes. OR swapped with new data presented in EV4.

We appreciate the suggestion and have moved Fig 3 D/E to EV Fig 3 and moved new EV data to main Figure 3.

P 9" marked by impaired oxidative phosphorylation" = decreased resting OCR, coupled respiration and maximal respiration. Do they have the data for activity per mito to present the ratio. I.e TMRM/mitotracker/spy. It seems that there is perhaps the same mito activity per mitomass, just lower mito mass.

Since the MitoTracker experiments and the flux assays have been performed with different samples, we cannot correlate these data, unfortunately.

The new data does not all need to go in supplements. Consider putting all key findings in main figures.

We have integrated several key findings from extended view figures into the main figures as suggested.

Referee #3 (Comments on Novelty/Model System for Author):

The study uses an appropriate and technically sound approach, combining patient samples with multiple omics platforms to obtain a comprehensive characterization, followed by in vitro assays that provide mechanistic insights. The integration of clinical material with experimental validation enhances the robustness of the findings and supports the adequacy of the model. However, the presentation of some data could be improved for clarity. For instance, volcano plots would benefit from including metabolite or enzyme names and indicating the omics technique used (e.g., proteomics, metabolomics) to facilitate interpretation. Similarly, the analysis of the RNA-seq dataset would be strengthened by a more systematic exploration of differentially regulated pathways and processes, beyond the focus on oxidative stress-related genes.

These findings have clear medical relevance, as they provide insights into how postoperative immune suppression may be metabolically mediated. Understanding this mechanism opens potential avenues for therapeutic interventions to preserve T cell function in the postsurgical setting, which could have direct clinical impact on patient outcomes.

As the authors have performed and included the additional experiments suggested in my initial review, I consider that the manuscript is now suitable for publication pending minor corrections. Most of these relate to editing and presentation. In particular, figure panels, labels, and graphs are generally too small, which makes interpretation difficult, and the extensive use of acronyms further complicates figure readability. Addressing these issues will substantially improve the accessibility of the manuscript.

We thank the reviewer for his valuable suggestions. We have now implemented a more comprehensive GO pathway analysis of RNA seq data to enable a more systematic exploration of regulated pathways and processes (Fig EV 3 H). Also, pathway enrichment analysis of metabolomic data has been expanded (Fig 2F/G).

We have comprehensively increased the size of all figure panels, and also increased the size of labels and graphs within the figure panels to improve the readability.

Referee #3 (Remarks for Author):

Thank you for taking into consideration my suggestions and comments. I only have a few remaining points for clarification and minor corrections:

Figure 1: Please clarify whether the parameters shown were measured in freshly isolated CTLs or in cells restimulated (e.g., with CD3/CD28 beads and/or PMA/ionomycin). This information is essential for data interpretation.

CTL have been stimulated using CD3/CD28 beads for mRNA/ protein secretion analysis. Only for intracellular flow cytometry, CTL have been additionally restimulated using PMA/Ionomycin. In addition to the methods section, we have added this information also to the figure legends to enable an easier interpretation.

Page 36, Figure 4f: The exemplary immunoblot of OXPHOS complexes in CTLs should be described more precisely. Please indicate that the blot shows representative proteins from the different OXPHOS complexes, rather than the intact complexes themselves. Ideally, specify in the figure legend which exact protein(s) were detected. Please, also correct this along the manuscript.

We thank the referee for this valuable remark. Indeed, this total OXPHOS Human WB Antibody Cocktail detects the following subunits of the respective OXPHOS complexes:

Complex I: subunit NDUFB8

Complex II: subunit 30kDa (SDHB)

Complex III: subunit Core 2 (UQCRC2)

Complex IV: subunit II (COXII)

Complex V: ATP synthase subunit alpha (ATP5A).

Accordingly, we changed this throughout the manuscript and in the respective figure legends

Figure EV4F: The sentence should be corrected from "compromised membrane integrity" to "compromised mitochondrial membrane potential integrity" for accuracy.

The sentence was corrected.

Figure EV4G: The data on impaired mitochondrial respiratory function are shown without the number of patients analyzed or a quantification. Please include both to strengthen this figure.

We apologize for this error. The OCR plot depicted is only an exemplary OCR plot from one patient. We have now added the missing quantification (basal, maximum and spare respiration) derived from three patients.

7th Oct 2025

Dear Prof. Kreth,

We are pleased to inform you that your manuscript is accepted for publication and is now being sent to our publisher to be included in the next available issue of EMBO Molecular Medicine.

Zeljko Durdevic
Senior Editor
EMBO Molecular Medicine
